# Smoothed Agnostic Learning of Halfspaces over the Hypercube

**Yiwen Kou**
Department of Computer Science, UCLA
Los Angeles, CA, US
evankou@cs.ucla.edu

**Raghu Meka**
Department of Computer Science, UCLA
Los Angeles, CA, US
raghum@cs.ucla.edu

## Abstract

Agnostic learning of Boolean halfspaces is a fundamental problem in computational learning theory, but it is known to be computationally hard even for weak learning. Recent work (Chandrasekaran et al., 2024) proposed smoothed analysis as a way to bypass such hardness, but existing frameworks rely on additive Gaussian perturbations, making them unsuitable for discrete domains. We introduce a new smoothed agnostic learning framework for Boolean inputs, where perturbations are modeled via random bit flips. This defines a natural discrete analogue of smoothed optimality generalizing the Gaussian case. Under strictly subexponential assumptions on the input distribution, we give an efficient algorithm for learning halfspaces in this model, with runtime and sample complexity $\widetilde{O}(n^{\mathrm{poly}(\frac{1}{\sigma\epsilon})})$. Previously, such algorithms were known only with strong structural assumptions for the discrete hypercube—for example, independent coordinates or symmetric distributions. Our result provides the first computationally efficient guarantee for smoothed agnostic learning of halfspaces over the Boolean hypercube, bridging the gap between worst-case intractability and practical learnability in discrete settings.

## 1 Introduction

Halfspaces, or linear threshold functions (LTFs), are one of the most fundamental concept classes in machine learning. In the realizable setting (Valiant, 1984), they are efficiently learnable by classical algorithms such as the Perceptron (Rosenblatt, 1958; Novikoff, 1963), Winnow (Littlestone, 1987), large-margin methods like Support Vector Machines (Cortes and Vapnik, 1995), or by linear programming. These methods exploit linear separability and can perform well even in the presence of irrelevant features.

In contrast, the agnostic learning framework (Haussler, 1992; Kearns et al., 1992), which allows for arbitrary label noise, poses significant algorithmic challenges. In this setting, the goal is to find a hypothesis that competes with the best in a concept class, without assuming that the data is linearly separable. However, agnostic learning of halfspaces is computationally hard in the worst case: even weak learning—achieving error marginally better than random guessing—is NP-hard under standard complexity assumptions, both in continuous domains (Feldman et al., 2009) and on the Boolean hypercube (Guruswami and Raghavendra, 2009).

To overcome these barriers, several restricted models have been studied. For example, under random classification noise (RCN), halfspaces remain learnable using modified Perceptron algorithms or linear programming (Blum et al., 1996; Cohen, 1997). Under Massart noise, where adversarial flips are bounded in probability, recent work has led to efficient learning algorithms (Awasthi et al., 2015; Diakonikolas et al., 2019, 2020, 2021). Other lines of work exploit structure in the input distribution: Kalai et al. (2008) gave improper agnostic learning algorithms under uniform, spherical,

or log-concave distributions by approximating halfspaces with low-degree polynomials, extending earlier Fourier-based methods (Linial et al., 1993).

A more recent and promising direction is based on smoothed analysis, which was introduced to explain the practical performance of algorithms that are worst-case hard (Spielman and Teng, 2001). In learning theory, Chandrasekaran et al. (2024) proposed a smoothed agnostic framework in which the learner competes with the best classifier under slight random perturbations of the inputs. This relaxation enables efficient algorithms for learning low-dimensional concepts, even when worst-case learning is intractable. However, their approach is tailored to continuous domains and relies on additive Gaussian noise, and hence is not suitable for discrete domains such as the Boolean hypercube.

**Our contribution.** We develop a discrete analogue of smoothed agnostic learning for Boolean concept classes over $\{\pm 1\}^n$, where additive Gaussian noise is ill-defined. Instead of perturbing examples in Euclidean space, we introduce bit-flipping noise: each input coordinate is independently flipped with probability $\sigma$. This gives rise to a new benchmark for learning that captures robustness to small discrete perturbations, interpolating between classical agnostic learning ($\sigma = 0$) and random guessing ($\sigma = 1/2$).

To formalize this, we begin by recalling the standard agnostic learning objective.

**Definition 1.1 (Agnostic Optimality)** *Let $\mathcal{X}$ be a domain and $\mathcal{F}$ be a class of functions $f : \mathcal{X} \to \{\pm 1\}$. Let $\mathcal{D}$ be a distribution over labeled examples $(\mathbf{x}, y) \in \mathcal{X} \times \{\pm 1\}$. The agnostic error of $\mathcal{F}$ under $\mathcal{D}$ is defined as*

$$\mathrm{opt} = \inf_{f \in \mathcal{F}} \mathbb{P}_{(\mathbf{x},y) \sim \mathcal{D}}[f(\mathbf{x}) \neq y].$$

We now define a smoothed variant of this benchmark, in which each input is perturbed before evaluation. This definition is general and does not assume any particular structure of the domain or distribution.

**Definition 1.2 (Smoothed Optimality)** *Let $\mathcal{X}$ be a domain and $\mathcal{F}$ be a class of functions $f : \mathcal{X} \to \{\pm 1\}$. Let $\mathcal{D}$ be a distribution over labeled examples $(\mathbf{x}, y) \in \mathcal{X} \times \{\pm 1\}$ and $\mathcal{P}_\sigma(\mathbf{x})$ be a perturbation distribution of $\mathbf{x}$ over $\mathcal{X}$. Define the smoothed agnostic error as:*

$$\mathrm{opt}_\sigma = \inf_{f \in \mathcal{F}} \mathbb{P}_{(\mathbf{x},y) \sim \mathcal{D}; \widetilde{\mathbf{x}} \sim \mathcal{P}_\sigma(\mathbf{x})}[f(\widetilde{\mathbf{x}}) \neq y].$$

In standard agnostic learning, the goal is to compete with $\mathrm{opt}$ under $\mathcal{D}$. Following Chandrasekaran et al. (2024), we instead compete with $\mathrm{opt}_\sigma$ in Definition 1.2.

**Definition 1.3 (Smoothed Agnostic Learning)** *Fix $\epsilon, \sigma > 0$ and $\delta \in (0, 1)$. An algorithm $\mathcal{A}$ learns the class $\mathcal{F}$ in the $\sigma$-smoothed agnostic setting if, given i.i.d. samples from $\mathcal{D}$, it outputs a hypothesis $h : \mathcal{X} \to \{\pm 1\}$ such that with probability at least $1 - \delta$:*

$$\mathbb{P}_{(\mathbf{x},y) \sim \mathcal{D}}[h(\mathbf{x}) \neq y] \leq \mathrm{opt}_\sigma + \epsilon.$$

As an example, Chandrasekaran et al. (2024) study the case where $\mathcal{X} = \mathbb{R}^n$ and the perturbation distribution is additive Gaussian noise: $\mathcal{P}_\sigma(\mathbf{x}) = \mathcal{N}(\mathbf{x}, \sigma^2 \mathbf{I}_n)$. This formulation relies on the Euclidean structure of $\mathbb{R}^n$ and does not extend to discrete domains. In our setting, we consider $\mathcal{X} = \{\pm 1\}^n$ and define perturbations via random bit flips: $\mathcal{P}_\sigma(\mathbf{x}) = \mathbf{x} \odot \mathbf{z}$ where $\mathbf{z} \sim \mathcal{N}_\sigma$ is a product distribution with $z_i = -1$ with probability $\sigma$ and 1 otherwise. This defines a natural smoothed learning model for the Boolean hypercube that avoids embedding the domain into $\mathbb{R}^n$.

Under this framework, we show that halfspaces over the Boolean cube are efficiently learnable in the smoothed agnostic model under mild distributional assumptions. Our approach extends the classical $L_1$-polynomial regression framework to this smoothed setting. The key idea is that every Boolean halfspace, when composed with small random bit-flip perturbations, admits a low-degree polynomial approximation under the input distribution. To establish this, we analyze a smoothed version of the halfspace defined via a noise operator (Definition 3.2), and construct approximators using Berry–Esseen-type arguments combined with critical index analysis to handle irregular weight vectors. We obtain the following result:

**Theorem 1.4 (Subgaussian-Informal, see also Theorem 4.8)** *Let $\mathcal{D}$ be a distribution on $\{\pm 1\}^n \times \{\pm 1\}$ with sub-gaussian $\mathbf{x}$-marginal of variance proxy $\sigma_0^2$. There exists an algorithm that learns*

*the class of linear threshold functions in the $\sigma$-smoothed setting with $N = n^{\text{poly}(\sigma_0/\sigma\epsilon)} \log(1/\delta)$ samples and $\text{poly}(n, N)$ runtime.*

This is the first result that establishes efficient smoothed agnostic learning of halfspaces over the Boolean hypercube. While our algorithm is improper, it achieves strong generalization guarantees under natural distributions. Previously, such results for the hypercube were only known under very restricted distributions as discussed below.

## 2 Related Work

**Distributional Assumptions in Halfspace Learning:** It is well-understood that agnostically learning halfspaces is intractable in the worst case (Feldman et al., 2009; Guruswami and Raghavendra, 2009), even under relatively benign noise models (Diakonikolas et al., 2022). This has motivated a long line of *distribution-specific* algorithms that guarantee learnability by leveraging assumptions on the data distribution. Early work focused on uniform or product distributions, where powerful Fourier-analytic techniques yield low-degree approximations (Linial et al., 1993; Klivans et al., 2004a; Blais et al., 2010). Under the uniform hypercube distribution, halfspace concepts exhibit strong Fourier concentration and low noise sensitivity (O'Donnell, 2021), enabling efficient learning via low-degree polynomial approximation (Klivans et al., 2004a). This was extended to symmetric distributions in Wimmer (2010) and to arbitrary product distributions in Blais et al. (2010). However, beyond these there are very few general classes of distributions over they hypercube where halfspaces are agnostically learnable.

For continuous distributions, halfspaces were shown to be agnostically learnable under log-concave distributions by Kalai et al. (2008) and this was later extended to intersections and other functions of halfspaces in Kane et al. (2013). Much like the discrete setting, until the recent work of Chandrasekaran et al. (2024), most positive results required strong structural assumptions on the marginal distribution of the examples. This work introduced a new smoothed agnostic model which led to several new results for learning halfspaces and functions of halfspaces for a much broader class of distributions (e.g., sub-gaussian or sub-exponential densities). Our work continues this progression to very general distributions, but focuses on the Boolean domain and shows that only mild tail bounds (strictly sub-exponential) suffice for efficient learning in the smoothed setting.

**Noise Models and Smoothed Analysis:** In parallel to distributional assumptions on $X$, a complementary line of work has tackled label noise models and smoothed analysis. The classical noise models include random classification noise (RCN), where each label is independently flipped with some probability. Blum et al. (1996) gave the first polynomial-time algorithm for learning a halfspace under random classification noise, exploiting the fact that a halfspace's margin makes it relatively robust to independent label flips. A stronger noise model is the Massart noise model, which bounds the adversary by a flipping probability $\eta < 1/2$ on each example. Diakonikolas et al. (2020) gave an efficient algorithm for learning halfspaces with Massart noise over log-concave distributions. On the other hand, with adversarial (malicious) noise, learning halfspaces requires additional assumptions. Klivans et al. (2009) designed efficient algorithms for origin-centered halfspaces under malicious noise by assuming isotropic log-concave distribution and small noise rate. In smoothed analysis of learning, one assumes that either the data (Blum and Dunagan, 2002; Kane et al., 2013) or the target concept (Chandrasekaran et al., 2024) is randomly perturbed, so that pathological arrangements are avoided. Chandrasekaran et al. (2024) introduced a smoothed agnostic PAC model in $\mathbb{R}^d$ where the learner competes against the best classifier that is robust to slight Gaussian perturbations of examples. Our work can be seen as a Boolean analogue of this idea: rather than perturbing continuous inputs, we require the optimal halfspace to be stable under small random label flips.

## 3 Preliminaries

We review relevant definitions from Boolean function analysis that will allow us to define a discrete smoothing operator and justify using it in place of the original linear threshold function. We use definitions from the analysis of Boolean functions over product spaces, following the framework of Mossel et al. (2005). Let $(\Omega_1, \mu_1), \ldots, (\Omega_n, \mu_n)$ be finite probability spaces and let $(\Omega, \mu)$ denote their product. In our setting, we take $\Omega_i = \{\pm 1\}$ and define $\mu$ to be the product distribution $\mathcal{N}_\sigma$, where each coordinate is 1 with probability $1 - \sigma$ and $-1$ with probability $\sigma$, independently.

**Definition 3.1 ($\rho$-noisy copy)** *Given $\mathbf{x} \in \Omega$ and $\rho \in [0, 1]$, a $\rho$-noisy copy of $\mathbf{x}$ is a random vector $\mathbf{y} \sim \mathcal{N}_\rho(\mathbf{x})$, where each coordinate $y_i$ is independently set to $x_i$ with probability $\rho$ and to an independent draw from $\mu_i$ with probability $1 - \rho$.*

**Definition 3.2 (Noise operator $T_\rho$)** *For any function $f : \Omega \to \mathbb{R}$ and $\rho \in [0, 1]$, the noise operator $T_\rho$ is defined as*

$$(T_\rho f)(\mathbf{x}) = \mathbb{E}_{\mathbf{y} \sim \mathcal{N}_\rho(\mathbf{x})}[f(\mathbf{y})].$$

This definition generalizes the Bonami–Beckner operator (Kahn et al., 1988) when $\mu$ is the uniform distribution on the hypercube. Intuitively, $T_\rho f$ is a smoothed version of $f$, computed by averaging $f$ over a neighborhood of $\mathbf{x}$ with geometric decay controlled by $\rho$. In particular, $T_1 f = f$, and as $\rho$ decreases from 1, $T_\rho f$ suppresses high-frequency components of $f$. This operator will be used as our main tool for constructing smoothed approximations to Boolean threshold functions.

**Definition 3.3 (Noise stability and noise sensitivity)** *For any $f : \Omega \to \mathbb{R}$, the noise stability at parameter $\rho$ is defined as*

$$S_\rho(f) = \langle f, T_\rho f \rangle_\mu.$$

*If $f : \Omega \to \{\pm 1\}$, the noise sensitivity at parameter $\delta \in [0, 1]$ is given by*

$$NS_\delta(f) = \frac{1}{2} - \frac{1}{2} S_{1-\delta}(f) = \mathbb{P}_{\mathbf{x} \sim \mu, \mathbf{y} \sim \mathcal{N}_{1-\delta}(\mathbf{x})}[f(\mathbf{x}) \neq f(\mathbf{y})].$$

Equivalently, $NS_\delta(f) = \mathrm{Pr}_{x,y}[f(x) \neq f(y)]$ where $x$ and $y$ have Hamming correlation $1 - 2\delta$. This quantity captures the robustness of $f$ to small input perturbations.

It is well-known that natural Boolean functions with low total influence or low-degree Fourier concentration exhibit low noise sensitivity. In particular, linear threshold functions are noise-stable under both uniform (Peres, 2004) and general product distributions (Blais et al., 2010). The following lemma bounds the noise sensitivity of halfspaces over arbitrary product spaces.

**Lemma 3.4 (Theorem 3.2 in Blais et al. (2010))** *Let $f : \Omega \to \{\pm 1\}$ is a linear threshold function, where the domain $\Omega = \Omega_1 \times \cdots \times \Omega_n$ has the product distribution $\mu = \mu_1 \times \cdots \times \mu_n$. Then $NS_\delta(f) \leq \frac{5}{4}\sqrt{\delta}$.*

This bound implies that for $\rho = 1 - \delta$ close to 1, the smoothed function $T_\rho f$ closely approximates the original threshold function $f$. This justifies our strategy of working with $T_\rho f$ instead of $f$ in the smoothed learning setting: any learner that performs well on $T_\rho f$ will, up to a small error, also succeed on $f$.

**Notation.** We use small boldface characters for vectors and capital bold characters for matrices. We use $[d]$ to denote the set $\{1, 2, \cdots, d\}$. For a vector $\mathbf{x} \in \mathbb{R}^d$ and $i \in [d]$, $x_i$ denotes the $i$-th coordinate of $\mathbf{x}$, and $\|\mathbf{x}\|_2 := \sqrt{\sum_{i=1}^d x_i^2}$ the $\ell_2$ norm of $\mathbf{x}$. For $\mathbf{x}, \mathbf{y} \in \mathbb{R}^d$, we use $\langle \mathbf{x}, \mathbf{y} \rangle = \sum_{i=1}^d x_i y_i$ as the inner product between them and $\mathbf{x} \odot \mathbf{y} = (x_1 y_1, \cdots, x_d y_d)$ as the Hadamard product between them. We use $\mathbf{1}\{\mathcal{E}\}$ to be the indicator function of some event $\mathcal{E}$. For $(\mathbf{x}, y)$ distributed according to $\mathcal{D}$, we denote $\mathcal{D}_\mathbf{x}$ to be the marginal distribution of $\mathbf{x}$.

## 4 Technical Overview

In this section, we outline the main steps of our analysis. Our approach follows a reduction-based strategy: we reduce smoothed agnostic learning of Boolean halfspaces to the problem of approximating a smoothed halfspace by a low-degree polynomial, which can then be learned via $L_1$ regression (Section 4.1). We begin by replacing the original target $f_\mathbf{x}(\mathbf{z}) = f(\mathbf{x} \odot \mathbf{z})$ with a smoothed surrogate $T_{1-\rho} f_\mathbf{x}(\mathbf{z})$ (Definition 3.2), facilitating approximation by low-degree polynomials (Section 4.2).

To handle the biased distribution arising from noise perturbation, we introduce a rerandomization and conditioning trick that rewrites each bit as a mixture involving uniform random variables. This allows us to express the smoothed function as a conditional expectation over uniformly random inputs, making it amenable to quantitative central-limit theorems (Berry–Esseen estimates; Section 4.3). We then use a case analysis facilitated by a decomposition of the weight vector (Section 4.4):

1. If a small number of large coordinates (the "head") dominate, the halfspace's output is primarily determined by those coordinates, and we can approximate the function directly.

2. Otherwise, the remaining "tail" is *regular*, and we apply the Berry–Esseen theorem to approximate the Boolean sum by a Gaussian. This reduces the problem to the continuous setting, where we leverage Gaussian-based techniques (the density ratio method from Chandrasekaran et al. (2024)) to construct low-degree polynomial approximations.

Together, these ingredients yield an efficient smoothed learner for Boolean halfspaces under strictly sub-exponential input distributions.

## 4.1 High-Level Approach via $L_1$ egression

Our starting point is the $L_1$-polynomial regression method for agnostic learning. In particular, Kalai et al. (2008) established a powerful reduction from agnostic learnability to low-degree polynomial approximation.

---

**Algorithm 1** $L_1$ Polynomial Regression Algorithm

---

**Input:** Sample $S = \{(x^1, y^1), \ldots, (x^N, y^N)\}$, degree bound $d$
 1: Find polynomial $p$ of degree $\leq d$ to minimize

$$\frac{1}{N} \sum_{j=1}^{N} |p(x^j) - y^j|.$$

   (This can be done by expanding examples to include all monomials of degree $\leq d$ and then performing $L_1$ linear regression.)
 2: Output hypothesis $h(x) = \mathrm{sign}(p(x) - t)$, where $t \in [-1, 1]$ is chosen to minimize the classification error on $S$.

---

**Theorem 4.1 (Theorem 5 in Kalai et al. (2008))** *Suppose* $\min_{\deg(p) \leq d} \mathbb{E}_{\mathcal{D}_{\mathbf{x}}}[|p(\mathbf{x}) - c(\mathbf{x})|] \leq \epsilon$ *for some degree $d$ and any $c$ in the concept class $\mathcal{C}$. Then, for $h$ output by the degree-$d$ $L_1$ polynomial regression algorithm with $N = \mathrm{poly}(n^d/\epsilon)$ examples,* $\mathbb{E}_{S \sim \mathcal{D}^N}[\mathbb{P}_{(\mathbf{x}, y) \sim \mathcal{D}}[h(\mathbf{x}) \neq y]] \leq \mathrm{opt} + \epsilon$, *where* $\mathrm{opt} = \min_{f \in \mathcal{C}} \mathbb{P}_{(\mathbf{x}, y) \sim \mathcal{D}}[f(\mathbf{x}) \neq y]$. *If we repeat the algorithm $r = O(\log(1/\delta)/\epsilon)$ times with fresh examples each, and let $h$ be the hypothesis with lowest error on an independent test set of size $O(\log(1/\delta)/\epsilon^2)$, then with probability at least $1 - \delta$,* $\mathbb{P}_{(\mathbf{x}, y) \sim \mathcal{D}}[h(\mathbf{x}) \neq y] \leq \mathrm{opt} + \epsilon$.

Theorem 4.1 says that if the target function $f$ can be approximated in $L_1$ by a low-degree polynomial $p$ with error at most $\epsilon$, then one can efficiently learn $f$ to misclassification error $\mathrm{opt} + \epsilon$, where $\mathrm{opt}$ is the Bayes-optimal error rate under distribution $\mathcal{D}$. Once such a polynomial is shown to exist, Theorem 4.1 implies a computationally efficient learning algorithm with sample complexity $N = \mathrm{poly}(n^d/\epsilon) \log(1/\delta)$.

## 4.2 Smoothed Learning as Non-Worst-Case Approximation

The challenge is that an arbitrary halfspace $f(\mathbf{x}) = \mathrm{sign}(\langle \mathbf{w}, \mathbf{x} \rangle - \theta)$ might not be well-approximated by any low-degree polynomial over worst-case input distributions. Following Chandrasekaran et al. (2024), we view smoothed learning as a form of non-worst-case approximation. In this smoothed agnostic setting, the learner's "effective" target concept is the mapping $(\mathbf{x}, \mathbf{z}) \mapsto f(\mathbf{x} \odot \mathbf{z})$, where $\mathbf{z} \in \{\pm 1\}^n$ is a random noise vector independent of $\mathbf{x}$ with $\sigma$ close to 0 meaning only a tiny fraction of bits are flipped on average. We extend the $L_1$-regression reduction to handle this scenario. In particular, we prove an analogue of Kalai et al. (2008)'s result tailored to the smoothed model:

**Theorem 4.2** *Suppose* $\min_{\deg(p_\mathbf{z}) \leq d} \mathbb{E}_{\mathbf{z} \sim \mathcal{D}_\sigma, \mathbf{x} \sim \mathcal{D}_{\mathbf{x}}}[|p_\mathbf{z}(\mathbf{x}) - f(\mathbf{x} \odot \mathbf{z})|] \leq \epsilon$ *for some degree $d$ and any halfspace $f$, where $\mathcal{D}_{\mathbf{x}}$ is any distribution on $\{\pm 1\}^n$. Then, for $h$ output by the degree-$d$ $L_1$ polynomial regression algorithm with $N = \mathrm{poly}(n^d/\epsilon)$ examples,* $\mathbb{E}_{S \sim \mathcal{D}^N}[\mathbb{P}_{(\mathbf{x}, y) \sim \mathcal{D}}[h(\mathbf{x}) \neq y]] \leq \mathrm{opt}_\sigma + \epsilon$. *If we repeat the algorithm $r = O(\log(1/\delta)/\epsilon)$ times with fresh examples each, and let $h$ be the hypothesis with lowest error on an independent test set of size $O(\log(1/\delta)/\epsilon^2)$, then with probability at least $1 - \delta$,* $\mathbb{P}_{(\mathbf{x}, y) \sim \mathcal{D}}[h(\mathbf{x}) \neq y] \leq \mathrm{opt}_\sigma + \epsilon$.

After this reduction, our task reduces to a purely approximation-theoretic problem: we need to construct, for each noise vector $\mathbf{z}$, a polynomial $p_{\mathbf{z}}(\mathbf{x})$ in the variable $\mathbf{x}$ such that the expected $L_1$ error over the smoothing process remains small:

$$\mathbb{E}_{\mathbf{z}\sim\mathcal{N}_\sigma,\mathbf{x}\sim D_{\mathbf{x}}}\big[|p_{\mathbf{z}}(\mathbf{x})-f(\mathbf{x}\odot\mathbf{z})|\big]\leq\epsilon.$$

To achieve this, we treat the smoothing noise $\mathbf{z}$ and consider $\mathbf{x}$ as a fixed parameter. This reduces the problem to approximating the function $f_{\mathbf{x}}(\mathbf{z})=f(\mathbf{x}\odot\mathbf{z})$. We replace $f_{\mathbf{x}}(\mathbf{z})$ with its smooth approximation by applying the generalized Bonami-Beckner operator (Definition 3.2) on $\mathbf{z}$:

$$T_{1-\rho}f_{\mathbf{x}}(\mathbf{z})=\mathbb{E}_{\mathbf{y}\sim\mathcal{N}_{1-\rho}(\mathbf{z})}[f_{\mathbf{x}}(\mathbf{y})].$$

Applying Lemma 3.4 with $\rho=O(\epsilon^2)$, we obtain:

$$\mathbb{E}_{\mathbf{z}\sim\mathcal{N}_\sigma}[|T_{1-\rho}f_{\mathbf{x}}(\mathbf{z})-f_{\mathbf{x}}(\mathbf{z})|]\leq\epsilon.$$

Therefore, if we can find a low-degree polynomial that approximates $T_{1-\rho}f_{\mathbf{x}}(\mathbf{z})$ well in $L_1$, that polynomial will also succeed in approximating $f_{\mathbf{x}}(\mathbf{z})$. The remainder of our technical approach will be devoted to constructing such a polynomial approximator for the smoothed halfspace $T_{1-\rho}f_{\mathbf{x}}(\mathbf{z})$.

## 4.3 From Biased to Uniform Distribution on the Hypercube

To construct low-degree polynomial approximations, we analyze the noise-smoothed function $T_{1-\rho}f_{\mathbf{x}}$. Recall that for a fixed input $\mathbf{x}$, we define $f_{\mathbf{x}}(\mathbf{z})=f(\mathbf{x}\odot\mathbf{z})$, and suppose $f(\cdot)=\mathrm{sign}(\langle\mathbf{w},\cdot\rangle-\theta)$. Then we have:

$$T_{1-\rho}f_{\mathbf{x}}(\mathbf{z})=\mathbb{E}_{\mathbf{y}\sim\mathcal{N}_{1-\rho}(\mathbf{z})}[\mathrm{sign}(\langle\mathbf{w}\odot\mathbf{x},\mathbf{y}\rangle-\theta)]=\mathbb{E}_{\mathbf{y}\sim\mathcal{N}_{1-\rho}(\mathbf{z})}[\mathrm{sign}(\langle\mathbf{u},\mathbf{y}\rangle-\theta)],$$

where we define $\mathbf{u}=\mathbf{w}\odot\mathbf{x}$.

Here, $\mathbf{z}\sim\mathcal{N}_\sigma$ denotes a product distribution over $\{\pm1\}^n$ where each bit $z_i$ is 1 with probability $1-\sigma$ and $-1$ with probability $\sigma$. The vector $\mathbf{y}$ is a $(1-\rho)$-noisy copy of $\mathbf{z}$ (Definition 3.1) with probability $1-\rho$, $y_i=z_i$; otherwise, $y_i$ is redrawn independently from $\mathcal{N}_\sigma$ with probability $\rho$. Therefore, $\mathbf{y}\sim\mathcal{N}_\sigma$, correlated with $\mathbf{z}$, follows a biased distribution on the hypercube. To facilitate polynomial approximation, we aim to reduce this to a form where the randomness comes from a uniform distribution. To achieve this, we introduce a rerandomization trick that rewrites each coordinate $y_i$ as:

$$y_i=(1-l_i)z_i+l_i\tau_i=(1-l_i)z_i+l_i(1-m_i)+l_im_i\epsilon_i,$$

where

$$l_i=\begin{cases}1\text{ w.p. }\rho\\0\text{ w.p. }1-\rho\end{cases},\tau_i=\begin{cases}1\text{ w.p. }1-\sigma\\-1\text{ w.p. }\sigma\end{cases},m_i=\begin{cases}1\text{ w.p. }2\sigma\\0\text{ w.p. }1-2\sigma\end{cases},$$

with $\epsilon_i$ being a Radmacher random variable (uniform over $\{\pm1\}$).

This decomposition captures the full noise process: $l_i$ is an indicator that determines whether the coordinate is kept as $z_i$ (with probability $1-\rho$) or resampled as $\tau_i\sim(\mathcal{N}_\sigma)_i$ (with probability $\rho$). The variable $m_i$ is then used to rerandomize $\tau_i$, since $\tau_i$ can be viewed as taking the value 1 with probability $1-2\sigma$ (when $m_i=0$) or a uniform random bit $\epsilon_i$ with probability $2\sigma$ (when $m_i=1$).

A key benefit is that, conditional on $\mathbf{l}$ and $\mathbf{m}$, the random component $\boldsymbol{\epsilon}$ follows the uniform distribution on $\{\pm1\}^n$. We now condition on $(\mathbf{l},\mathbf{m})$ and express the smoothed function as:

$$T_{1-\rho}f_{\mathbf{x}}(\mathbf{z})=\mathbb{E}_{\mathbf{l},\mathbf{m}}\Big[\mathbb{E}_{\mathbf{y}}[\mathrm{sign}(\langle\mathbf{u},\mathbf{y}\rangle-\theta)|\mathbf{l},\mathbf{m}]\Big]=\mathbb{E}_{\mathbf{l},\mathbf{m}}\Big[\mathbb{E}_{\boldsymbol{\epsilon}}[\mathrm{sign}(\langle\mathbf{u},\mathbf{l}\odot\mathbf{m}\odot\boldsymbol{\epsilon}\rangle+b-\theta)|\mathbf{l},\mathbf{m}]\Big],$$

where $b$ is a deterministic shift depending on the coordinates fixed by $\mathbf{l},\mathbf{m}$.

Given that $\boldsymbol{\epsilon}$ is uniform distribution on hypercube, the inner sum behaves like a sum of independent $\{\pm1\}$ random variables. Under mild regularity condition (Definition 4.3) on the weight vector $\mathbf{u}$, we can apply the Berry–Esseen Theorem to approximate this inner distribution by a Gaussian. Specifically, we approximate:

$$\langle\mathbf{u},\mathbf{l}\odot\mathbf{m}\odot\boldsymbol{\epsilon}\rangle\approx\mathcal{N}(0,\|\mathbf{u}\odot\mathbf{l}\odot\mathbf{m}\|_2^2).\tag{4.1}$$

Substituting into the earlier expression yields the Gaussian-smoothed approximation:

$$\widetilde{T_{1-\rho}f_{\mathbf{x}}}(\mathbf{z})=\mathbb{E}_{\mathbf{l},\mathbf{m}}\Big[\mathbb{E}_{s\sim\mathcal{N}(0,\|\mathbf{u}\odot\mathbf{l}\odot\mathbf{m}\|_2^2)}[\mathrm{sign}(s+b-\theta)|\mathbf{l},\mathbf{m}]\Big]$$

This reduces our setting to the Gaussian noise model analyzed in Chandrasekaran et al. (2024) for which efficient low-degree polynomial approximations are known. In particular, the density ratio method developed in that work can be applied to approximate $\widetilde{T_{1-\rho}f_{\mathbf{x}}}(\mathbf{z})$ with a small $L_1$ error.

## 4.4 Handling Irregularity via Critical Index Analysis

Recall that the approximation in (4.1) relies on the Berry–Esseen Theorem, which introduces a uniform approximation error of $O\big(\big(\frac{\|\mathbf{u}\odot\mathbf{l}\odot\mathbf{m}\|_3}{\|\mathbf{u}\odot\mathbf{l}\odot\mathbf{m}\|_2}\big)^3\big)$ for the cumulative density function. This can be further bounded by $O\big(\frac{\|\mathbf{u}\odot\mathbf{l}\odot\mathbf{m}\|_\infty}{\|\mathbf{u}\odot\mathbf{l}\odot\mathbf{m}\|_2}\big)$. Note that each coordinate $l_i m_i$ is equal to 1 with probability $2\rho\sigma$ and 0 otherwise. By concentration, we have $\|\mathbf{u}\odot\mathbf{l}\odot\mathbf{m}\|_2 \approx (2\rho\sigma)^{1/2}\|\mathbf{u}\|_2$, so the approximation error becomes $O\big((\rho\sigma)^{-1/2}\frac{\|\mathbf{u}\|_\infty}{\|\mathbf{u}\|_2}\big)$. This motivates the following regularity condition:

**Definition 4.3 (regularity)** *For vector $\mathbf{w} \in \mathbb{R}^n$, $\mathbf{w}$ is $\alpha$-regular if $\|\mathbf{w}\|_\infty \leq \alpha \cdot \|\mathbf{w}\|_2$.*

Given this definition, we see that if $\mathbf{u}$ is $\alpha$-regular, then the approximation in (4.1) holds with $L_\infty$ error $O((\rho\sigma)^{-1/2}\alpha)$. Since $\mathbf{u} = \mathbf{w}\odot\mathbf{x}$ and $\mathbf{x} \in \{\pm 1\}^n$, the regularity of $\mathbf{u}$ is equivalent to that of $\mathbf{w}$. For such "good" (i.e., $\alpha$-regular with small $\alpha$) weight vectors $\mathbf{w}$, we can construct low-degree polynomial approximators by reducing to the Gaussian setting analyzed in Section 4.3 and Chandrasekaran et al. (2024).

However, we must also handle the "bad" or irregular cases, where $\langle \mathbf{u}, \mathbf{l}\odot\mathbf{m}\odot\boldsymbol{\epsilon}\rangle$ deviates significantly from Gaussian behavior. To deal with such irregular $\mathbf{w}$, we employ critical index analysis, a standard tool in the analysis of Boolean halfspaces (Servedio, 2006; Matulef et al., 2010; Diakonikolas et al., 2010; Meka and Zuckerman, 2010; O'Donnell and Servedio, 2011; Diakonikolas and Servedio, 2013).

**Definition 4.4 ($\alpha$-critical index)** *For $\mathbf{u} \in \mathbb{R}^n$, assume that $|u_1| \geq \cdots \geq |u_n|$. We define the $\alpha$-critical index $\ell(\alpha)$ of a halfspace $h(\mathbf{x}) = \text{sign}(\langle\mathbf{u},\mathbf{x}\rangle - \theta)$ as the smallest index $i \in [n]$ for which $|u_i| \leq \alpha \cdot \sigma_i$, where $\sigma_i := \sqrt{\sum_{j=i}^n u_j^2}$.*

Intuitively, the $\alpha$-critical index is the first index $i$ such that the tail weight vector $(u_i, \cdots, u_n)$ is $\alpha$-regular. Our earlier argument covers the case $i = 1$, where the entire vector is regular. Using this framework, we obtain the following structural result:

**Lemma 4.5 (Critical Index Decomposition)** *Without loss of generality, let $\mathbf{u} = \mathbf{w}\odot\mathbf{x}$ with entries sorted in non-increasing magnitude, i.e., $|u_1| \geq \cdots \geq |u_n|$. Suppose $\mathbf{x}$ follows a $(\alpha, \lambda)$-strictly sub-exponential distribution on $\{\pm 1\}^n$. For any fixed $\mathbf{z}$, there exists a threshold $K = K(\alpha,\epsilon) = O\big(\log(1+\lambda)/\alpha^2 + \log(1/\epsilon)\log(1/\alpha)/\rho\sigma\alpha^2\big)$ such that one of the following two conditions holds:*

*1. For some $H < K$, the tail vector $\mathbf{u}_T = (u_{H+1}, \cdots, u_n)$ is $\alpha$-regular, where $\alpha$ is to be choosen later.*

*2. For $H = K$ and at least $1 - \epsilon$ fraction of $\mathbf{x}$, it holds that*
$$\mathbb{P}_{\mathbf{y}\sim\mathcal{N}_{1-\rho}(\mathbf{z})}[\text{sign}(\langle\mathbf{u}_H,\mathbf{y}_H\rangle + \langle\mathbf{u}_T,\mathbf{y}_T\rangle - \theta) \neq \text{sign}(\langle\mathbf{u}_H,\mathbf{y}_H\rangle - \theta)] \leq \epsilon, \qquad (4.2)$$
*where $\mathbf{u}_H := (u_1, \cdots, u_H)$.*

This lemma is proved by analyzing two cases, depending on whether the critical index $\ell(\alpha)$ satisfies $1 < \ell(\alpha) < K$, or $\ell(\alpha) \geq K$. In the former case, we set $H = \ell(\alpha) - 1$, and $\mathbf{u}_T$ is $\alpha$-regular. In the latter case, the head vector forms a sufficiently long geometrically decaying sequence $|u_1| \geq \cdots \geq |u_H|$ to ensure that the influence of the remaining tail vector $\mathbf{u}_T$ on the halfspace output is negligible. That is, with high probability, $\text{sign}(\langle\mathbf{u},\mathbf{y}\rangle - \theta) \approx \text{sign}(\langle\mathbf{u}_H,\mathbf{y}_H\rangle - \theta)$.

We now show how to construct low-degree polynomial approximators in both cases.

**Case 1:** When $\mathbf{u}_T$ is $\alpha$-regular, we condition on $\mathbf{y}_H$. For each fixed $\mathbf{y}_H$, the function becomes a regular halfspace in $\mathbf{y}_T$:
$$\widetilde{f}_{\mathbf{x}}(\mathbf{y}_T) = \text{sign}(\langle\mathbf{u}_T,\mathbf{y}_T\rangle - \widetilde{\theta}), \text{ where } \widetilde{\theta} = \theta - \langle\mathbf{u}_H,\mathbf{y}_H\rangle.$$
We apply the techniques of Chandrasekaran et al. (2024) to approximate this with a low-degree polynomial. One subtlety is that directly applying their construction leads to a degree polynomial in $|\widetilde{\theta}|/\|\mathbf{u}_T\|_2$. To address this, we use an indicator trick to define:
$$p_{\mathbf{y}_H}(\mathbf{x}) = \text{sign}(\langle\mathbf{u}_H,\mathbf{y}_H\rangle - \theta) \cdot \mathbb{1}\big(|\langle\mathbf{u}_H,\mathbf{y}_H\rangle - \theta| > C \cdot \|\mathbf{u}_T\|_2\big)$$
$$+ \widetilde{p}_{\mathbf{y}_H}(\mathbf{x}) \cdot \mathbb{1}\big(|\langle\mathbf{u}_H,\mathbf{y}_H\rangle - \theta| \leq C \cdot \|\mathbf{u}_T\|_2\big),$$

where $\widetilde{p}_{\mathbf{y}_H}(\mathbf{x})$ can be constructed using the idea from Chandrasekaran et al. (2024) since $|\widetilde{\theta}|/\|\mathbf{u}_T\|_2$ is controlled. The indicator functions are low-degree polynomials of degree at most $H$, since they only depend on $H$ variables and any function $f : \{\pm 1\}^k \to \mathbb{R}$ can be represented by a degree at most $k$ multilinear polynomial.

**Case 2:** If the second condition of the lemma holds, we approximate $T_{1-\rho}f_{\mathbf{x}}(\mathbf{z})$ directly using $\text{sign}(\langle \mathbf{u}_H, \mathbf{y}_H \rangle - \theta)$. Since this depends only on the first $H$ coordinates, it can be exactly represented as a polynomial of degree at most $H$.

In either case, we obtain a low-degree polynomial approximator for the smoothed function $T_{1-\rho}f_{\mathbf{x}}(\mathbf{z})$.

## 4.5 Results

Using this framework, we establish the following approximation bound:

**Definition 4.6 (Strictly Sub-exponential Distributions)** *A distribution $\mathcal{D}$ on $\mathbb{R}^d$ is $(\alpha, \lambda)$-strictly sub-exponential if for all $\|\mathbf{v}\|_2 = 1$, $\mathbb{P}_{\mathbf{x} \sim \mathcal{D}}[|\langle \mathbf{x}, \mathbf{v} \rangle| > t] \leq 2 \cdot e^{-(t/\lambda)^{1+\alpha}}$.*

**Lemma 4.7** *Fix $\epsilon > 0$ and a sufficiently large universal constant $C > 0$. Let $\mathcal{D}$ be a $(\alpha, \lambda)$-strictly sub-exponential distribution on $\{\pm 1\}^n$. Let $f : \{\pm 1\}^n \to \{\pm 1\}$ be a linear threshold function. There exists a family of polynomials $p_{\mathbf{z}}$ parameterized by $\mathbf{z}$ of degree at most $O\left(\left(C\sigma^{-\frac{1}{2}}\lambda \log(1/\epsilon)/\epsilon\right)^{6(1+\frac{1}{\alpha})^3}\right)$ such that $\mathbb{E}_{\mathbf{z} \sim \mathcal{N}_\sigma} \mathbb{E}_{\mathbf{x} \sim D}[|p_{\mathbf{z}}(\mathbf{x}) - f_{\mathbf{x}}(\mathbf{z})|]$ is at most $\epsilon$.*

Given the polynomial approximation and the degree upper bound, one can directly run $L_1$ polynomial regression (Algorithm 1) as stated in Theorem 4.2. We now can get our main theorem for strictly sub-exponential distributions.

**Theorem 4.8** *Let $\mathcal{D}$ be a distribution on $\{\pm 1\}^n \times \{\pm 1\}$ such that the marginal distribution is $(\alpha, \lambda)$-strictly sub-exponential. There exists an algorithm that draws $N = n^{\text{poly}((\lambda/\sigma\epsilon)^{(1+1/\alpha)^3})} \log(1/\delta)$ samples, runs in time $\text{poly}(n, N)$, and computes a hypothesis $h(\mathbf{x})$ such that, with probability at least $1 - \delta$, it holds that $\mathbb{P}_{(\mathbf{x},y) \sim \mathcal{D}}[y \neq h(\mathbf{x})] \leq \text{opt}_\sigma + \epsilon$.*

Our main theorem shows that any Boolean halfspace on $\{\pm 1\}^n$ can be learned agnostically in the smoothed model under strictly sub-exponential input distributions. This result holds in a general and challenging setting where prior techniques fail, and it achieves efficient runtime and sample complexity. Table 1 compares our guarantees with the most relevant prior works. Conceptually, our contributions extend the scope of agnostic halfspace learning in two fundamental directions:

**Relaxing distributional assumptions via smoothed optimality:** A key technical contribution is our use of a smoothed benchmark $\text{opt}_\sigma$ (Definition 1.3) instead of the worst-case error $\text{opt}$ (Definition 1.1), enabling learning under substantially weaker distributional assumptions. In particular, we show that halfspaces remain efficiently learnable under general strictly sub-exponential marginals, which is a significant relaxation compared to the strong structural assumptions required in earlier work. For example, the Fourier-based techniques of Klivans et al. (2004b); Kalai et al. (2008); Blais et al. (2010) exploit spectral concentration under uniform or product distributions to obtain low-degree polynomial approximations. To go beyond the product setting, Wimmer (2010) generalized previous techniques to symmetric group to handle permutation-invariant distributions. However, these methods break down when the input has more dependencies or heavier tails. In contrast, our approach succeeds under strictly sub-exponential marginals by combining bit-flip smoothing with a critical index decomposition and Berry–Esseen approximation, enabling polynomial approximation without requiring coordinate independence or permutation invariant structure.

**Extending the smoothed learning framework to the Boolean hypercube:** A second core contribution is our extension of smoothed agnostic learning to the Boolean domain $\{\pm 1\}^n$, where additive Gaussian perturbations used in prior smoothed models are not well-defined. In continuous domains, several tools, including Gaussian surface area bounds (Klivans et al., 2008), log-concave concentration inequalities (Kane et al., 2013), and Gaussian smoothing combined with density ratio techniques (Chandrasekaran et al., 2024), enable efficient agnostic learning of halfspaces. However, none of these directly apply to the hypercube. Our analysis circumvents this barrier by performing a case analysis based on the critical index of the weight vector: either a small number of head coordinates

Table 1: Comparison of agnostic learning of halfspaces. We ignore the polynomial logarithmic factors in $1/\delta$.

| Work | Domain | Distribution | Bench. | Smooth | Complexity |
|---|---|---|---|---|---|
| Kalai et al. (2008) | $\{\pm 1\}^n$ | Uniform | opt | None | $n^{O(\frac{1}{\epsilon^4})}$ |
| Kalai et al. (2008) | $S^{n-1}$ | Uniform | opt | None | $n^{O(\frac{1}{\epsilon^4})}$ |
| Kalai et al. (2008) | $\mathbb{R}^n$ | Log-concave | opt | None | $n^{O(d(\epsilon))}$ |
| Klivans et al. (2008) | $\mathbb{R}^n$ | Gaussian | opt | None | $n^{O(\frac{1}{\epsilon^4})}$ |
| Wimmer (2010) | $[B]^n$ | Perm-Inv | opt | None | $n^{O(\frac{1}{\epsilon^4})}$ |
| Kane et al. (2013) | $\mathbb{R}^n$ | Sub-exp | $\text{opt}_\sigma$ | Input noise | $n^{\exp(\frac{(\log\log(\frac{1}{\sigma\epsilon}))^{\widetilde{O}(1)}}{\sigma^4\epsilon^4})}$ |
| Chandrasekaran et al. (2024) | $\mathbb{R}^n$ | Strictly Sub-exp | $\text{opt}_\sigma$ | Concept noise | $n^{\text{poly}((\frac{\lambda}{\sigma\epsilon})^{(1+\frac{1}{\alpha})^3})}$ |
| **Ours** | $\{\pm 1\}^n$ | Strictly Sub-exp | $\text{opt}_\sigma$ | Concept noise | $n^{\text{poly}((\frac{\lambda}{\sigma\epsilon})^{(1+\frac{1}{\alpha})^3})}$ |

dominate and effectively determine the output, or the remaining tail is regular, allowing us to invoke the Berry–Esseen theorem to approximate the Boolean tail sum by a Gaussian, thereby enabling the use of continuous tools developed in prior work (Chandrasekaran et al., 2024).

## 5   Conclusion and Open Problems

In this work, we extended the smoothed agnostic learning framework to the Boolean hypercube, and demonstrated that halfspaces are efficiently learnable with respect to a broad class of input distributions (strictly sub-exponential marginals). Our approach combines tools from smoothing analysis, conditional polynomial approximation, and critical index decomposition to construct low-degree polynomial approximators in a discrete setting where standard analytic techniques are not applicable. By competing with a smoothed benchmark $\text{opt}_\sigma$, our guarantee circumvents known hardness results for agnostic learning over the hypercube, while matching the sample and runtime complexity of prior work in continuous domains.

Our current techniques apply only to single halfspaces, and the polynomial degree and runtime degrade as the smoothing parameter $\sigma$ becomes small. In addition, our analysis requires strictly sub-exponential tail assumptions, and it remains unclear whether comparable guarantees are achievable under weaker conditions. Our results also suggest a potential link to agnostic learning under smoothed input distributions, analogous to the Gaussian framework in continuous domains (Kalai and Teng, 2008; Kalai et al., 2009; Kane et al., 2013; Chandrasekaran et al., 2024). Formalizing this connection in the Boolean setting appears subtle, due to the lack of Euclidean geometry and the discrete nature of bit-flip noise, and we leave it as an intriguing direction for future work.

An important open question is whether these techniques can be extended to intersections of multiple halfspaces. While our framework theoretically supports such generalizations under smoothed optimality, a major technical challenge arises in adapting critical index analysis to this setting. For a single halfspace, sorting the coordinates of the weight vector by magnitude plays a central role in identifying regular and irregular components. However, in the case of multiple halfspaces, each weight vector may induce a different ordering over coordinates, making it difficult to define a unified notion of "head" and "tail" variables. As a result, applying a shared conditioning or decomposition strategy becomes nontrivial. Developing new structural insights or approximation techniques that can handle this multi-directional irregularity remains an open problem.

More broadly, this raises the question of how far the smoothed learning framework can be pushed. Can it yield efficient algorithms for learning other complex Boolean concept classes (e.g., DNF formulas, decision lists, or polynomial threshold functions of higher degree) under heavy-tailed distributions? Can it be made adaptive to unknown noise levels or to distributions that do not satisfy strict tail bounds? We leave these questions for future work.

## Acknowledgments and Disclosure of Funding

We sincerely thank the anonymous reviewers for their helpful comments. The authors acknowledge support in part from the National Science Foundation under Award CCF-2217033 (EnCORE: Institute for Emerging CORE Methods in Data Science).

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

## A  Bonami-Beckner Operator Approximation

We show that for any linear threshold function $f : \{\pm 1\}^n \to \{\pm 1\}$ the approximation error $L_1$ of the operator $T_{1-\rho}f$ to $f$ can be upper bounded by $O(\sqrt{\rho})$.

**Lemma A.1** *For any linear threshold function $f : \{\pm 1\}^n \to \{\pm 1\}$ and $\sigma, \rho \in [0, 1]$, it holds that*

$$\mathbb{E}_{\mathbf{z} \sim \mathcal{N}_\sigma}[|T_{1-\rho}f(\mathbf{z}) - f(\mathbf{z})|] \leq \frac{5}{2}\sqrt{\rho}.$$

**Proof** By triangle inequality and special case of Lemma 3.4 when $\Omega = \{\pm 1\}^n$ and $\mu = \mathcal{N}_\sigma$, we have

$$
\begin{aligned}
\mathbb{E}_{\mathbf{z} \sim \mathcal{N}_\sigma}[|T_{1-\rho}f(\mathbf{z}) - f(\mathbf{z})|] &= \mathbb{E}_{\mathbf{z} \sim \mathcal{N}_\sigma}\left[\left|\mathbb{E}_{\mathbf{y} \sim \mathcal{N}_{1-\rho}(\mathbf{z})}[f(\mathbf{y})] - f(\mathbf{z})\right|\right] \\
&\leq \mathbb{E}_{\mathbf{z} \sim \mathcal{N}_\sigma}\left[\mathbb{E}_{\mathbf{y} \sim \mathcal{N}_{1-\rho}(\mathbf{z})}[|f(\mathbf{y}) - f(\mathbf{z})|]\right] \\
&= 2\mathbb{E}_{\mathbf{z} \sim \mathcal{N}_\sigma, \mathbf{y} \sim \mathcal{N}_{1-\rho}(\mathbf{z})}\left[\mathbb{1}[f(\mathbf{y}) \neq f(\mathbf{z})]\right] \\
&\leq 2NS_\rho(f) \\
&\leq \frac{5}{2}\sqrt{\rho}.
\end{aligned}
$$

■

Therefore, choosing $\rho = O(\epsilon^2)$ makes this error at most $\epsilon/2$.

## B  Polynomial Approximation for $T_{1-\rho}f_{\mathbf{x}}(\mathbf{z})$

We now approximate $T_{1-\rho}f_{\mathbf{x}}(\mathbf{z})$ using a polynomial for the more general class of strictly sub-exponential distributions.

**Definition B.1 (Strictly Sub-exponential Distributions)** *A distribution $\mathcal{D}$ on $\mathbb{R}^d$ is $(\alpha, \lambda)$-strictly sub-exponential if for all $\|\mathbf{v}\|_2 = 1$, $\mathbb{P}_{\mathbf{x} \sim \mathcal{D}}[|\langle \mathbf{x}, \mathbf{v} \rangle| > t] \leq 2 \cdot e^{-(t/\lambda)^{1+\alpha}}$.*

Our main goal in this section is to prove the following polynomial approximation result in Lemma 4.7. Suppose $f(\mathbf{x}) = \text{sign}(\langle \mathbf{w}, \mathbf{x} \rangle - \theta)$. Denote $\mathbf{w} \odot \mathbf{x}$ as $\mathbf{u}$. Without loss of generality, suppose that $|u_1| \geq |u_2| \geq \cdots \geq |u_n|$. Then, we have

$$T_{1-\rho}f_{\mathbf{x}}(\mathbf{z}) = \mathbb{E}_{\mathbf{y} \sim \mathcal{N}_{1-\rho}(\mathbf{z})}[\text{sign}(\langle \mathbf{w} \odot \mathbf{x}, \mathbf{y} \rangle - \theta)] = \mathbb{E}_{\mathbf{y} \sim \mathcal{N}_{1-\rho}(\mathbf{z})}[\text{sign}(\langle \mathbf{u}, \mathbf{y} \rangle - \theta)].$$

To obtain a polynomial approximation of $T_\rho f_{\mathbf{x}}(\mathbf{z})$, we first prove Lemma 4.5.

**Proof** Let

$$
\begin{aligned}
L(\alpha, \epsilon) &= \lceil \log(1/\epsilon)/\rho\sigma \rceil \cdot \lceil (4/\alpha^2)\log(1/\alpha) \rceil + 2\log(C/\alpha)/\alpha^2 \\
&= O\big(\log(1/\epsilon)\log(1/\alpha)/\rho\sigma\alpha^2\big) + O\big(\log(\max\{\lambda, 1\}\log^{\max\{\frac{1}{2}, \frac{1}{1+\alpha}\}}(1/\epsilon)/\alpha)/\alpha^2\big) \\
&= O\big(\log(1/\epsilon)\log(1/\alpha)/\rho\sigma\alpha^2\big) + O\big(\log(\max\{\lambda, 1\}\log(1/\epsilon)/\alpha)/\alpha^2\big) \\
&= O\big(\log(1+\lambda)/\alpha^2 + \log(1/\epsilon)\log(1/\alpha)/\rho\sigma\alpha^2\big),
\end{aligned}
$$

where $C = (1 - 2\rho\sigma) \cdot \lambda \log^{\frac{1}{1+\alpha}}(2/\epsilon) + \sqrt{2\log(2/\epsilon)}$.

If $\alpha$-critical index $\ell(\alpha) < L(\alpha, \epsilon)$, then the first condition holds by taking $H = \ell(\alpha) - 1$. If the $\alpha$-critical index $\ell(\epsilon) \geq L(\alpha, \epsilon)$, we will show that the second condition holds.

By Lemma 5.5 in Diakonikolas et al. (2010), there exist a set of nicely separated coordinates $G = \{i_1, i_2, \cdots, i_t\} \subseteq H$ where $i_1 < i_2 < \cdots < i_t$ and $i_{k+1} - i_k = \lceil 4\log(1/\alpha)/\alpha^2 \rceil$ such that $|u_{i_{k+1}}| \leq |u_{i_k}|/3$ for any $k \in [t-1]$. Then by Claim 5.7 in Diakonikolas et al. (2010), for any two points $\mathbf{x}_1 \neq \mathbf{x}_2 \in \{\pm 1\}^t$, we have $|\langle \mathbf{u}_G, \mathbf{x}_1 \rangle - \langle \mathbf{u}_G, \mathbf{x}_2 \rangle| \geq |u_{i_t}|$. Take $t = \lceil \log(1/\epsilon)/\rho\sigma \rceil$. For any

fixed assignment to the variables in $H \setminus G$, we have

$$\mathbb{P}_{\mathbf{y}}\left[\left|\sum_{i \in H} u_i y_i - \theta\right| \leq \frac{|u_{i_t}|}{4}\right]$$

$$= \mathbb{P}_{\mathbf{y}}\left[\sum_{i \in G} u_i y_i \in \left[\theta - \sum_{i \in H \setminus G} u_i y_i - \frac{|u_{i_t}|}{4}, \theta - \sum_{i \in H \setminus G} u_i y_i + \frac{|u_{i_t}|}{4}\right]\right]$$

$$\leq \max_{\mathbf{x}_1 \in \{\pm 1\}^t} \mathbb{P}_{\mathbf{y}_G}[\mathbf{y}_G = \mathbf{x}_1]$$

$$\leq (1 - \rho\sigma)^t \leq e^{-\rho\sigma t} \leq \epsilon,$$

where the second inequality is because there's at most one point in an interval of length $|u_{i_t}|$ given that $\langle \mathbf{u}_G, \mathbf{x}_1 \rangle$ are well-separated. The third inequality is because

$$\max\{\mathbb{P}[y_i = 1], \mathbb{P}[y_i = -1]\} \leq \max\{1 - \rho\sigma, \rho\sigma\} = 1 - \rho\sigma, \qquad \text{if } z_i = 1,$$
$$\max\{\mathbb{P}[y_i = 1], \mathbb{P}[y_i = -1]\} \leq \max\{\rho(1 - \sigma), 1 - \rho(1 - \sigma)\} \leq 1 - \rho\sigma, \quad \text{if } z_i = -1.$$

By our choice of $L(\alpha, \epsilon), t, i_t$, we have

$$L(\alpha, \epsilon) - i_t \geq L(\alpha, \epsilon) - \lceil \log(1/\epsilon)/\rho\sigma \rceil \cdot \lceil (4/\alpha^2) \log(1/\alpha) \rceil \geq 2\log(C/\alpha)/\alpha^2.$$

By applying Lemma 5.5 in Diakonikolas et al. (2010), we have

$$\|\mathbf{u}_T\|_2 = \sigma_T \leq (\sqrt{1 - \alpha^2})^{L(\alpha,\epsilon) - i_t} \cdot |u_{i_t}|/\alpha \leq C^{-1} \cdot |u_{i_t}|.$$

Therefore, by Lemma C.1, for at least $1 - \epsilon$ fraction of $\mathbf{x}$, it holds with probability at least $1 - \epsilon$ of $\mathbf{y}$ that

$$|\langle \mathbf{u}_H, \mathbf{y}_H \rangle - \theta| = \left|\sum_{i \in H} u_i y_i - \theta\right| \geq \frac{|u_{i_t}|}{4} \geq \frac{C \cdot \|\mathbf{u}_T\|_2}{4} \geq \frac{|\langle \mathbf{u}_T, \mathbf{y}_T \rangle|}{4}.$$

Then, it follows that with probability at least $1 - \epsilon$ of $\mathbf{y}$ we have

$$\text{sign}\left(\langle \mathbf{u}_H, \mathbf{y}_H \rangle + \langle \mathbf{u}_T, \mathbf{y}_T \rangle - \theta\right) = \text{sign}\left(\langle \mathbf{u}_H, \mathbf{y}_H \rangle - \theta\right).$$

∎

Now we are ready to construct our polynomial. We consider the two cases in Lemma 4.5 separately.

**Case 1:** If the weight vector $\mathbf{w}$ of the LTF falls into the second case of Lemma 4.5, notice that $\text{sign}(\langle \mathbf{u}_H, \mathbf{y}_H \rangle - \theta) = \text{sign}(\langle \mathbf{w}_H \odot \mathbf{x}_H, \mathbf{y}_H \rangle - \theta)$ can be represented as a polynomial $p_{\mathbf{y}}(\mathbf{x})$ of degree at most $H = K$ since only $H$ coordinates of $\mathbf{x}$ are relevant. In this case, we take our final polynomial as

$$p_{\mathbf{z}}(\mathbf{x}) = \mathbb{E}_{\mathbf{y} \sim \mathcal{N}_{1-\rho}(\mathbf{z})}[p_{\mathbf{y}}(\mathbf{x})] = \mathbb{E}_{\mathbf{y} \sim \mathcal{N}_{1-\rho}(\mathbf{z})}[\text{sign}(\langle \mathbf{w}_H \odot \mathbf{x}_H, \mathbf{y}_H \rangle - \theta)].$$

Let $\Delta(\mathbf{x})$ be defined as the error term $\mathbb{E}_{\mathbf{z} \sim \mathcal{N}_\sigma}[|p_{\mathbf{z}}(\mathbf{x}) - T_{1-\rho} f_{\mathbf{x}}(\mathbf{z})|]$. We have that for at least $1 - \epsilon$ fraction of $\mathbf{x}$ it holds that

$$\Delta(\mathbf{x}) = \mathbb{E}_{\mathbf{z} \sim \mathcal{N}_\sigma}\left[\left|\mathbb{E}_{\mathbf{y} \sim \mathcal{N}_{1-\rho}(\mathbf{z})}[\text{sign}(\langle \mathbf{w}_H \odot \mathbf{x}_H, \mathbf{y}_H \rangle - \theta)] - \mathbb{E}_{\mathbf{y} \sim \mathcal{N}_{1-\rho}(\mathbf{z})}[\text{sign}(\langle \mathbf{w} \odot \mathbf{x}, \mathbf{y} \rangle - \theta)]\right|\right]$$

$$\leq \mathbb{E}_{\mathbf{z} \sim \mathcal{N}_\sigma}\left[\mathbb{E}_{\mathbf{y} \sim \mathcal{N}_{1-\rho}(\mathbf{z})}\left[\left|\text{sign}(\langle \mathbf{w}_H \odot \mathbf{x}_H, \mathbf{y}_H \rangle - \theta) - \text{sign}(\langle \mathbf{w} \odot \mathbf{x}, \mathbf{y} \rangle - \theta)\right|\right]\right]$$

$$= 2\mathbb{E}_{\mathbf{z} \sim \mathcal{N}_\sigma}\left[\mathbb{P}_{\mathbf{y} \sim \mathcal{N}_{1-\rho}(\mathbf{z})}\left[\text{sign}(\langle \mathbf{w}_H \odot \mathbf{x}_H, \mathbf{y}_H \rangle - \theta) \neq \text{sign}(\langle \mathbf{w} \odot \mathbf{x}, \mathbf{y} \rangle - \theta)\right]\right]$$

$$\leq 2\epsilon.$$

It follows that the final $L_1$ approximation error

$$\mathbb{E}_{\mathbf{x} \sim \mathcal{D}_{\mathbf{x}}}\left[\mathbb{E}_{\mathbf{z} \sim \mathcal{N}_\sigma}[|p_{\mathbf{z}}(\mathbf{x}) - T_{1-\rho} f_{\mathbf{x}}(\mathbf{z})|]\right]$$

$$= \mathbb{E}_{\mathbf{x} \sim D_{\mathbf{x}}}[\Delta(\mathbf{x})]$$

$$= \mathbb{E}_{\mathbf{x} \sim D_{\mathbf{x}}}[\Delta(\mathbf{x})|\text{``good''} \mathbf{x}] \cdot \mathbb{P}_{\mathbf{x} \sim D_{\mathbf{x}}}[\text{``good''} \mathbf{x}] + \mathbb{E}_{\mathbf{x} \sim D_{\mathbf{x}}}[\Delta(\mathbf{x})|\text{``bad''} \mathbf{x}] \cdot \mathbb{P}_{\mathbf{x} \sim D_{\mathbf{x}}}[\text{``bad''} \mathbf{x}]$$

$$\leq 2\epsilon \cdot 1 + 2 \cdot \epsilon = 4\epsilon.$$

Here "good" $\mathbf{x}$ refers to the at least $1 - \epsilon$ fraction of $\mathbf{x}$ such that the approximation in (4.2) holds.

**Case 2:** If the weight vector $\mathbf{w}$ of the LTF falls into the first case of Lemma 4.5, we consider the following approximation

$$p_{\mathbf{y}_H}(\mathbf{x}) = \text{sign}(\langle \mathbf{u}_H, \mathbf{y}_H \rangle - \theta) \cdot \mathbb{1}\left(|\langle \mathbf{u}_H, \mathbf{y}_H \rangle - \theta| > C \cdot \|\mathbf{u}_T\|_2\right)$$
$$+ \widetilde{p}_{\mathbf{y}_H}(\mathbf{x}) \cdot \mathbb{1}\left(|\langle \mathbf{u}_H, \mathbf{y}_H \rangle - \theta| \le C \cdot \|\mathbf{u}_T\|_2\right),$$

where $C = (1 - 2\rho\sigma) \cdot \lambda \log^{\frac{1}{1+\alpha}}(2/\epsilon) + \sqrt{2\log(2/\epsilon)}$ and $\widetilde{p}_{\mathbf{y}_H}(\mathbf{x})$ will be choosen later as (B.5). In this case, we take our final polynomial as

$$p_{\mathbf{z}}(\mathbf{x}) = \mathbb{E}_{\mathbf{y}_H \sim \mathcal{N}_{1-\rho}(\mathbf{z})|_H}[p_{\mathbf{y}_H}(\mathbf{x})].$$

Let $\Delta(\mathbf{x})$ be defined as the $L_1$ error term $\mathbb{E}_{\mathbf{z} \sim \mathcal{N}_\sigma}[|p_{\mathbf{z}}(\mathbf{x}) - T_{1-\rho}f_{\mathbf{x}}(\mathbf{z})|]$. For notation simplicity, we denote $\{\mathbf{y}_H : |\langle \mathbf{u}_H, \mathbf{y}_H \rangle - \theta| > C \cdot \|\mathbf{u}_T\|_2\}$ as event $\mathcal{E}$. Then, we have

$\Delta(\mathbf{x})$

$= \mathbb{E}_{\mathbf{z} \sim \mathcal{N}_\sigma}\left[\left|\mathbb{E}_{\mathbf{y} \sim \mathcal{N}_{1-\rho}(\mathbf{z})}[p_{\mathbf{y}}(\mathbf{x})] - \mathbb{E}_{\mathbf{y} \sim \mathcal{N}_{1-\rho}(\mathbf{z})}[f_{\mathbf{x}}(\mathbf{y})]\right|\right]$

$= \mathbb{E}_{\mathbf{z} \sim \mathcal{N}_\sigma}\left[\left|\mathbb{E}_{\mathbf{y} \sim \mathcal{N}_{1-\rho}(\mathbf{z})}\left[(p_{\mathbf{y}}(\mathbf{x}) - f_{\mathbf{x}}(\mathbf{y})) \cdot \mathbb{1}[\mathcal{E}]\right] + \mathbb{E}_{\mathbf{y} \sim \mathcal{N}_{1-\rho}(\mathbf{z})}\left[(p_{\mathbf{y}}(\mathbf{x}) - f_{\mathbf{x}}(\mathbf{y})) \cdot \mathbb{1}[\mathcal{E}^c]\right]\right|\right]$

$\le \mathbb{E}_{\mathbf{z} \sim \mathcal{N}_\sigma}\left[\left|\mathbb{E}_{\mathbf{y} \sim \mathcal{N}_{1-\rho}(\mathbf{z})}\left[(p_{\mathbf{y}}(\mathbf{x}) - f_{\mathbf{x}}(\mathbf{y})) \cdot \mathbb{1}[\mathcal{E}]\right]\right| + \left|\mathbb{E}_{\mathbf{y} \sim \mathcal{N}_{1-\rho}(\mathbf{z})}\left[(p_{\mathbf{y}}(\mathbf{x}) - f_{\mathbf{x}}(\mathbf{y})) \cdot \mathbb{1}[\mathcal{E}^c]\right]\right|\right]$

$= \underbrace{\mathbb{E}_{\mathbf{z} \sim \mathcal{N}_\sigma}\left[\left|\mathbb{E}_{\mathbf{y} \sim \mathcal{N}_{1-\rho}(\mathbf{z})}\left[\left(\text{sign}(\langle \mathbf{u}, \mathbf{y} \rangle - \theta) - \text{sign}(\langle \mathbf{u}_H, \mathbf{y}_H \rangle - \theta)\right) \cdot \mathbb{1}[\mathcal{E}]\right]\right|\right]}_{\Delta_1(\mathbf{x})}$

$+ \underbrace{\mathbb{E}_{\mathbf{z} \sim \mathcal{N}_\sigma}\left[\left|\mathbb{E}_{\mathbf{y} \sim \mathcal{N}_{1-\rho}(\mathbf{z})}\left[\left(\text{sign}(\langle \mathbf{u}, \mathbf{y} \rangle - \theta) - \widetilde{p}_{\mathbf{y}}(\mathbf{x})\right) \cdot \mathbb{1}[\mathcal{E}^c]\right]\right|\right]}_{\Delta_2(\mathbf{x})}$

Notice that by Lemma C.1, for at least $1 - \epsilon$ fraction of $\mathbf{x}$, $|\langle \mathbf{u}_T, \mathbf{y}_T \rangle| \le C \cdot \|\mathbf{u}_T\|_2$ holds for at least $1 - \epsilon$ fraction of $\mathbf{y}$. For such $\mathbf{x}$ and $\mathbf{y}$, under event $\mathcal{E}$, we have

$$|\langle \mathbf{u}_T, \mathbf{y}_T \rangle| \le C \cdot \|\mathbf{u}_T\|_2 < |\langle \mathbf{u}_H, \mathbf{y}_H \rangle - \theta|,$$

then it follows that

$$\text{sign}(\langle \mathbf{u}, \mathbf{y} \rangle - \theta) \cdot \mathbb{1}[\mathcal{E}] = \text{sign}(\langle \mathbf{u}_H, \mathbf{y}_H \rangle + \langle \mathbf{u}_T, \mathbf{y}_T \rangle - \theta) \cdot \mathbb{1}[\mathcal{E}] = \text{sign}(\langle \mathbf{u}_H, \mathbf{y}_H \rangle - \theta) \cdot \mathbb{1}[\mathcal{E}].$$

Then, for at least $1 - \epsilon$ fraction of $\mathbf{x}$ we have

$\left|\mathbb{E}_{\mathbf{y} \sim \mathcal{N}_{1-\rho}(\mathbf{z})}\left[\left(\text{sign}(\langle \mathbf{u}, \mathbf{y} \rangle - \theta) - \text{sign}(\langle \mathbf{u}_H, \mathbf{y}_H \rangle - \theta)\right) \cdot \mathbb{1}[\mathcal{E}]\right]\right|$

$\le \left|\mathbb{E}_{\mathbf{y} \sim \mathcal{N}_{1-\rho}(\mathbf{z})}\left[\left(\text{sign}(\langle \mathbf{u}, \mathbf{y} \rangle - \theta) - \text{sign}(\langle \mathbf{u}_H, \mathbf{y}_H \rangle - \theta)\right) \cdot \mathbb{1}[\mathcal{E}]\big|\text{"good" }\mathbf{y}\right]\right|$

$\quad\quad \cdot \mathbb{P}_{\mathbf{y} \sim \mathcal{N}_{1-\rho}(\mathbf{z})}[\text{"good" }\mathbf{y}]$

$+ \left|\mathbb{E}_{\mathbf{y} \sim \mathcal{N}_{1-\rho}(\mathbf{z})}\left[\left(\text{sign}(\langle \mathbf{u}, \mathbf{y} \rangle - \theta) - \text{sign}(\langle \mathbf{u}_H, \mathbf{y}_H \rangle - \theta)\right) \cdot \mathbb{1}[\mathcal{E}]\big|\text{"bad" }\mathbf{y}\right]\right|$

$\quad\quad \cdot \mathbb{P}_{\mathbf{y} \sim \mathcal{N}_{1-\rho}(\mathbf{z})}[\text{"bad" }\mathbf{y}]$

$\le 0 + 2 \cdot \epsilon = 2\epsilon.$

Here "good" $\mathbf{y}$ refers to the at least $1 - \epsilon$ fraction of $\mathbf{y}$ such that $|\langle \mathbf{u}_T, \mathbf{y}_T \rangle| \le C \cdot \|\mathbf{u}_T\|_2$ holds. Therefore, we have

$$\mathbb{E}_{\mathbf{x} \sim \mathcal{D}_{\mathbf{x}}}[\Delta_1(\mathbf{x})] = \mathbb{E}_{\mathbf{x} \sim \mathcal{D}_{\mathbf{x}}}[\Delta_1(\mathbf{x})|\text{"good" }\mathbf{x}] \cdot \mathbb{P}_{\mathbf{x} \sim \mathcal{D}_{\mathbf{x}}}[\text{"good" }\mathbf{x}]$$
$$+ \mathbb{E}_{\mathbf{x} \sim \mathcal{D}_{\mathbf{x}}}[\Delta_1(\mathbf{x})|\text{"bad" }\mathbf{x}] \cdot \mathbb{P}_{\mathbf{x} \sim \mathcal{D}_{\mathbf{x}}}[\text{"bad" }\mathbf{x}]$$
$$\le \mathbb{E}_{\mathbf{x} \sim \mathcal{D}_{\mathbf{x}}}[\Delta_1(\mathbf{x})|\text{"good" }\mathbf{x}] + 2 \cdot \epsilon$$
$$\le 2\epsilon + 2\epsilon = 4\epsilon.$$

Here "good" $\mathbf{x}$ refers to the at least $1 - \epsilon$ fraction of $\mathbf{x}$ such that $\mathbb{P}_{\mathbf{y}}[|\langle \mathbf{u}_T, \mathbf{y}_T \rangle| \le C \cdot \|\mathbf{u}_T\|_2] \ge 1 - \epsilon$ holds.

Next we consider bounding $\Delta_2(\mathbf{x})$ by constructing proper low-degree polynomial $\widetilde{p}_{\mathbf{y}}(\mathbf{x})$ as follows. Recall that $\mathbf{y} \sim \mathcal{N}_{1-\rho}(\mathbf{z})$ where $y_i = z_i$ with probability $1 - \rho$ and $y_i$ randomly drawn from $\mathcal{N}_\sigma$ with

probability $\rho$, we use the following rerandomization trick for random vector $\mathbf{y}_T$: for each coordinate of $\mathbf{y}_T$, let

$$\left.\begin{array}{l} y_i = (1 - l_i)z_i + l_i\tau_i \\ \tau_i = (1 - m_i) + m_i\epsilon_i \end{array}\right\} \implies y_i = (1 - l_i)z_i + l_i(1 - m_i) + l_i m_i \epsilon_i, \tag{B.1}$$

where

$$l_i = \begin{cases} 1 \text{ w.p. } \rho \\ 0 \text{ w.p. } 1 - \rho \end{cases}, \tau_i = \begin{cases} 1 \text{ w.p. } 1 - \sigma \\ -1 \text{ w.p. } \sigma \end{cases}, m_i = \begin{cases} 1 \text{ w.p. } 2\sigma \\ 0 \text{ w.p. } 1 - 2\sigma \end{cases},$$

and $\epsilon_i$ is a Radmacher random variable. Let random variable $A = \langle \mathbf{u}, \mathbf{y} \rangle - \theta$. Then by (B.1) we have

$$\begin{aligned} A &= \langle \mathbf{u}, \mathbf{y} \rangle - \theta \\ &= \langle \mathbf{u}_H, \mathbf{y}_H \rangle + \langle \mathbf{u}_T, \mathbf{y}_T \rangle - \theta \\ &= \langle \mathbf{u}_H, \mathbf{y}_H \rangle + \langle \mathbf{u}_T, (\mathbf{1}_T - \mathbf{l}_T) \odot \mathbf{z}_T \rangle + \langle \mathbf{u}_T, \mathbf{l}_T \odot (\mathbf{1}_T - \mathbf{m}_T) \rangle - \theta + \langle \mathbf{u}_T, \mathbf{l}_T \odot \mathbf{m}_T \odot \boldsymbol{\epsilon}_T \rangle \\ &= \underbrace{\langle \mathbf{u}_H, \mathbf{y}_H \rangle + \langle \mathbf{u}_T, \mathbf{v}_T \rangle - \theta}_{:=b} + \underbrace{\langle \mathbf{u}_T, \mathbf{l}_T \odot \mathbf{m}_T \odot \boldsymbol{\epsilon}_T \rangle}_{:=B}. \end{aligned}$$

where $\mathbf{v}_T := (\mathbf{1}_T - \mathbf{l}_T) \odot \mathbf{z}_T + \mathbf{l}_T \odot (\mathbf{1}_T - \mathbf{m}_T)$. Then we have

$$\begin{aligned} \widetilde{T_{1-\rho} f_{\mathbf{x}}}(\mathbf{z}) &:= \mathbb{E}_{\mathbf{y} \sim \mathcal{N}_{1-\rho}(\mathbf{z})} \big[ \operatorname{sign}(\langle \mathbf{u}, \mathbf{y} \rangle - \theta) \cdot \mathbb{1}[\mathbf{y}_H \in \mathcal{E}^c] \big] \\ &= \mathbb{E}_{\mathbf{y}_H, A} [\operatorname{sign}(A) \cdot \mathbb{1}[\mathbf{y}_H \in \mathcal{E}^c]] \\ &= \mathbb{E}_{\mathbf{y}_H, \mathbf{l}_T, \mathbf{m}_T} \big[ \mathbb{E}_A[\operatorname{sign}(A)|\mathbf{y}_H, \mathbf{l}_T, \mathbf{m}_T] \cdot \mathbb{1}[\mathbf{y}_H \in \mathcal{E}^c] \big] \\ &= \mathbb{E}_{\mathbf{y}_H, \mathbf{l}_T, \mathbf{m}_T} \big[ \big(1 - 2\mathbb{P}_A[A \le 0|\mathbf{y}_H, \mathbf{l}_T, \mathbf{m}_T]\big) \cdot \mathbb{1}[\mathbf{y}_H \in \mathcal{E}^c] \big] \\ &= \mathbb{E}_{\mathbf{y}_H, \mathbf{l}_T, \mathbf{m}_T} \big[ \big(1 - 2\mathbb{P}_B[B \le -b|\mathbf{y}_H, \mathbf{l}_T, \mathbf{m}_T]\big) \cdot \mathbb{1}[\mathbf{y}_H \in \mathcal{E}^c] \big], \end{aligned}$$

and

$$\begin{aligned} &\Delta_2(\mathbf{x}) \\ &= \mathbb{E}_{\mathbf{z} \sim \mathcal{N}_\sigma} \Big[ \Big| \mathbb{E}_{\mathbf{y} \sim \mathcal{N}_{1-\rho}(\mathbf{z})} \big[ \operatorname{sign}(\langle \mathbf{u}, \mathbf{y} \rangle - \theta) \cdot \mathbf{1}[\mathbf{y}_H \in \mathcal{E}^c] \big] - \mathbb{E}_{\mathbf{y} \sim \mathcal{N}_{1-\rho}(\mathbf{z})} \big[ \widetilde{p}_{\mathbf{y}}(\mathbf{x}) \cdot \mathbf{1}[\mathbf{y}_H \in \mathcal{E}^c] \big] \Big| \Big] \\ &= \mathbb{E}_{\mathbf{z} \sim \mathcal{N}_\sigma} \Big[ \Big| \widetilde{T_{1-\rho} f_{\mathbf{x}}}(\mathbf{z}) - \mathbb{E}_{\mathbf{y} \sim \mathcal{N}_{1-\rho}(\mathbf{z})} \big[ \widetilde{p}_{\mathbf{y}}(\mathbf{x}) \cdot \mathbf{1}[\mathbf{y}_H \in \mathcal{E}^c] \big] \Big| \Big]. \end{aligned}$$

By Theorem C.3, we have

$$\sup_{x \in \mathbb{R}} \left| \mathbb{P} \left[ \frac{B}{\|\mathbf{u}_T \odot \mathbf{l}_T \odot \mathbf{m}_T\|_2} \le x \Big| \mathbf{l}_T, \mathbf{m}_T \right] - \Phi(x) \right| \le C' \cdot \left( \frac{\|\mathbf{u}_T \odot \mathbf{l}_T \odot \mathbf{m}_T\|_3}{\|\mathbf{u}_T \odot \mathbf{l}_T \odot \mathbf{m}_T\|_2} \right)^3.$$

where $C'$ is a constant. Therefore, we have

$$\begin{aligned} &\left| \widetilde{T_{1-\rho} f_{\mathbf{x}}}(\mathbf{z}) - \mathbb{E}_{\mathbf{y}_H, \mathbf{l}_T, \mathbf{m}_T} \left[ \left( 1 - 2\Phi \left( \frac{-b}{\|\mathbf{u}_T \odot \mathbf{l}_T \odot \mathbf{m}_T\|_2} \right) \right) \cdot \mathbb{1}[\mathbf{y}_H \in \mathcal{E}^c] \right] \right| \\ &\le 2\mathbb{E}_{\mathbf{y}_H, \mathbf{l}_T, \mathbf{m}_T} \left[ \left| \mathbb{P}[B \le -b|\mathbf{y}_H, \mathbf{l}_T, \mathbf{m}_T] - \Phi \left( \frac{-b}{\|\mathbf{u}_T \odot \mathbf{l}_T \odot \mathbf{m}_T\|_2} \right) \right| \cdot \mathbb{1}[\mathbf{y}_H \in \mathcal{E}^c] \right] \\ &\le 2C' \cdot \mathbb{E}_{\mathbf{y}_H, \mathbf{l}_T, \mathbf{m}_T} \left[ \left( \frac{\|\mathbf{u}_T \odot \mathbf{l}_T \odot \mathbf{m}_T\|_3}{\|\mathbf{u}_T \odot \mathbf{l}_T \odot \mathbf{m}_T\|_2} \right)^3 \cdot \mathbb{1}[\mathbf{y}_H \in \mathcal{E}^c] \right] \\ &\le 2C' \cdot \mathbb{E}_{\mathbf{y}_H, \mathbf{l}_T, \mathbf{m}_T} \left[ \frac{\|\mathbf{u}_T \odot \mathbf{l}_T \odot \mathbf{m}_T\|_\infty}{\|\mathbf{u}_T \odot \mathbf{l}_T \odot \mathbf{m}_T\|_2} \cdot \mathbb{1}[\mathbf{y}_H \in \mathcal{E}^c] \right]. \end{aligned}$$

Notice that in this case $\mathbf{u}_T$ is $\alpha$-regular and $l_i m_i$ takes 1 with probability $2\rho\sigma$ and 0 with probability $1 - 2\rho\sigma$, then by Lemma C.2, for at least $1 - \epsilon$ fraction of $(\mathbf{l}_T, \mathbf{m}_T)$ it holds that

$$\|\mathbf{u}_T\|_\infty \le \alpha \cdot \|\mathbf{u}_T\|_2 \le (\rho\sigma)^{-\frac{1}{2}} \cdot \alpha \cdot \|\mathbf{u}_T \odot \mathbf{l}_T \odot \mathbf{m}_T\|_2, \tag{B.2}$$

as long as the condition $\alpha \le \rho\sigma/\sqrt{\log(1/\epsilon)/2}$ holds.

Notice that

$$1 - 2\Phi\left(\frac{-b}{\|\mathbf{u}_T \odot \mathbf{l}_T \odot \mathbf{m}_T\|_2}\right) = \mathbb{E}_{X \sim \mathcal{N}(b, \|\mathbf{u}_T \odot \mathbf{l}_T \odot \mathbf{m}_T\|_2^2)}[\text{sign}(X)|\mathbf{y}_H, \mathbf{l}_T, \mathbf{m}_T]$$

$$= \mathbb{E}_{X \sim \mathcal{N}(b, \|\mathbf{w}_T \odot \mathbf{l}_T \odot \mathbf{m}_T\|_2^2)}[\text{sign}(X)|\mathbf{y}_H, \mathbf{l}_T, \mathbf{m}_T],$$

let

$$\overline{T_\rho f_\mathbf{x}(\mathbf{z})} := \mathbb{E}_{\mathbf{y}_H, \mathbf{l}_T, \mathbf{m}_T}\Big[\mathbb{E}_{X \sim \mathcal{N}(b, \|\mathbf{w}_T \odot \mathbf{l}_T \odot \mathbf{m}_T\|_2^2)}[\text{sign}(X)|\mathbf{y}_H, \mathbf{l}_T, \mathbf{m}_T]$$

$$\cdot \mathbb{1}[(\mathbf{l}_T, \mathbf{m}_T) \in \mathcal{E}_1] \cdot \mathbb{1}[\mathbf{y}_H \in \mathcal{E}^c]\Big],$$

where $\mathcal{E}_1$ is the event regarding the randomness of $(\mathbf{l}_T, \mathbf{m}_T)$ such that (B.2) holds, then

$$\left|\widetilde{T_\rho f_\mathbf{x}(\mathbf{z})} - \overline{T_\rho f_\mathbf{x}(\mathbf{z})}\right|$$

$$\leq 2\mathbb{E}_{\mathbf{y}_H, \mathbf{l}_T, \mathbf{m}_T}\Bigg[\bigg|\mathbb{P}[B \leq -b|\mathbf{y}_H, \mathbf{l}_T, \mathbf{m}_T] - \Phi\bigg(-\frac{b}{\|\mathbf{u}_T \odot \mathbf{l}_T \odot \mathbf{m}_T\|_2}\bigg)\bigg|$$

$$\cdot \mathbb{1}[(\mathbf{l}_T, \mathbf{m}_T) \in \mathcal{E}_1] \cdot \mathbb{1}[\mathbf{y}_H \in \mathcal{E}^c]\Bigg]$$

$$+ \mathbb{E}_{\mathbf{y}_H, \mathbf{l}_T, \mathbf{m}_T}\Big[\big|1 - 2\mathbb{P}[B \leq -b|\mathbf{y}_H, \mathbf{l}_T, \mathbf{m}_T]\big| \cdot \mathbb{1}[(\mathbf{l}_T, \mathbf{m}_T) \in \mathcal{E}_1^c] \cdot \mathbb{1}[\mathbf{y}_H \in \mathcal{E}^c]\Big]$$

$$\leq 2C' \cdot \mathbb{E}_{\mathbf{y}_H, \mathbf{l}_T, \mathbf{m}_T}\Bigg[\frac{\|\mathbf{u}_T \odot \mathbf{l}_T \odot \mathbf{m}_T\|_\infty}{\|\mathbf{u}_T \odot \mathbf{l}_T \odot \mathbf{m}_T\|_2} \cdot \mathbb{1}[(\mathbf{l}_T, \mathbf{m}_T) \in \mathcal{E}_1] \cdot \mathbb{1}[\mathbf{y}_H \in \mathcal{E}^c]\Bigg]$$

$$+ \mathbb{E}_{\mathbf{y}_H, \mathbf{l}_T, \mathbf{m}_T}\Big[\mathbb{1}[(\mathbf{l}_T, \mathbf{m}_T) \in \mathcal{E}_1^c] \cdot \mathbb{1}[\mathbf{y}_H \in \mathcal{E}^c]\Big]$$

$$\leq 2C' \cdot (\rho\sigma)^{-\frac{1}{2}}\alpha + \epsilon$$

$$\leq 2C' \cdot (2\rho\sigma/\log(1/\epsilon))^{\frac{1}{2}} + \epsilon$$

$$\leq 2\epsilon,$$

as long as $\rho = O(\epsilon^2 \log(1/\epsilon)/\sigma)$. Then, we have

$$\mathbb{E}_{\mathbf{x} \sim \mathcal{D}_\mathbf{x}}\mathbb{E}_{\mathbf{z} \sim \mathcal{N}_\sigma}\Big[\big|\widetilde{T_\rho f_\mathbf{x}(\mathbf{z})} - \overline{T_\rho f_\mathbf{x}(\mathbf{z})}\big|\Big] \leq 2\epsilon.$$

Now we only need to consider polynomial approximation for $\overline{T_\rho f_\mathbf{x}(\mathbf{z})}$. We can recenter the expectation around zero as follows:

$$\mathbb{E}_{X \sim \mathcal{N}(b, \|\mathbf{w}_T \odot \mathbf{l}_T \odot \mathbf{m}_T\|_2^2)}[\text{sign}(X)|\mathbf{y}_H, \mathbf{l}_T, \mathbf{m}_T]$$

$$= \mathbb{E}_{s \sim \mathcal{N}(b/\|\mathbf{w}_T \odot \mathbf{l}_T \odot \mathbf{m}_T\|_2, 1)}[\text{sign}(\|\mathbf{w}_T \odot \mathbf{l}_T \odot \mathbf{u}_T\|_2 \cdot s)|\mathbf{y}_H, \mathbf{l}_T, \mathbf{m}_T]$$

$$= \mathbb{E}_{s \sim Q}\Bigg[\text{sign}(\|\mathbf{w}_T \odot \mathbf{l}_T \odot \mathbf{m}_T\|_2 \cdot s) \cdot \frac{\mathcal{N}(s; b/\|\mathbf{w}_T \odot \mathbf{l}_T \odot \mathbf{m}_T\|_2, 1)}{Q(s)}\bigg|\mathbf{y}_H, \mathbf{l}_T, \mathbf{m}_T\Bigg]$$

$$= \frac{e^{-\frac{b^2}{2\|\mathbf{w}_T \odot \mathbf{l}_T \odot \mathbf{m}_T\|_2^2}}}{\sqrt{2\pi}} \cdot \mathbb{E}_{s \sim Q}\Bigg[\text{sign}(\|\mathbf{w}_T \odot \mathbf{l}_T \odot \mathbf{m}_T\|_2 \cdot s) \cdot e^{-\frac{s^2}{2} - \log Q(s)}$$

$$\cdot e^{\frac{b \cdot s}{\|\mathbf{w}_T \odot \mathbf{l}_T \odot \mathbf{m}_T\|_2}}\bigg|\mathbf{y}_H, \mathbf{l}_T, \mathbf{m}_T\Bigg],$$

where $Q(s) = e^{-|s|}/2$.

For the simplicity of the following analysis, we define random variable

$$x := \frac{\langle \mathbf{u}_T, \mathbf{v}_T \rangle}{\|\mathbf{u}_T \odot \mathbf{v}_T\|_2} = \frac{\langle \mathbf{w}_T \odot \mathbf{x}_T, \mathbf{v}_T \rangle}{\|\mathbf{u}_T \odot \mathbf{v}_T\|_2} = \frac{\langle \mathbf{w}_T \odot \mathbf{v}_T, \mathbf{x}_T \rangle}{\|\mathbf{w}_T \odot \mathbf{v}_T\|_2}.$$

Since $\mathbf{x}$ is $(\alpha, \lambda)$-strictly sub-exponential, then $x$ satisfies the following concentration inequality:

$$\mathbb{P}_{\mathbf{x} \sim \mathcal{D}_\mathbf{x}}\big[|x| > t\big] \leq 2 \cdot e^{-(t/\lambda)^{1+\alpha}}.$$

Then, we can rewrite:

$$\frac{b}{\|\mathbf{w}_T \odot \mathbf{l}_T \odot \mathbf{m}_T\|_2} = \frac{(\langle \mathbf{u}_H, \mathbf{y}_H \rangle - \theta) + \langle \mathbf{u}_T, \mathbf{v}_T \rangle}{\|\mathbf{w}_T \odot \mathbf{l}_T \odot \mathbf{m}_T\|_2} = \frac{(\langle \mathbf{u}_H, \mathbf{y}_H \rangle - \theta) + \|\mathbf{w}_T \odot \mathbf{v}_T\|_2 \cdot x}{\|\mathbf{w}_T \odot \mathbf{l}_T \odot \mathbf{m}_T\|_2}.$$

For simplicity, denote

$$a = \frac{\|\mathbf{w}_T \odot \mathbf{v}_T\|_2}{\|\mathbf{w}_T \odot \mathbf{l}_T \odot \mathbf{m}_T\|_2}, \qquad c = \frac{\langle \mathbf{u}_H, \mathbf{y}_H \rangle - \theta}{\|\mathbf{w}_T \odot \mathbf{l}_T \odot \mathbf{m}_T\|_2}.$$

Notice that under condition $(\mathbf{l}_T, \mathbf{m}_T) \in \mathcal{E}_1$ and condition $\mathbf{y}_H \in \mathcal{E}^c$, we have

$$|c| = \frac{|\langle \mathbf{u}_H, \mathbf{y}_H \rangle - \theta|}{\|\mathbf{w}_T \odot \mathbf{l}_T \odot \mathbf{m}_T\|_2} \leq C \cdot \frac{\|\mathbf{u}_T\|_2}{\|\mathbf{w}_T \odot \mathbf{l}_T \odot \mathbf{m}_T\|_2} \leq C \cdot (\rho\sigma)^{-1/2} := c', \quad \text{(B.3)}$$

$$\begin{aligned}
|a| &\leq \frac{\|\mathbf{w}_T \odot (\mathbf{1}_T - \mathbf{l}_T) \odot \mathbf{z}_T\|_2 + \|\mathbf{w}_T \odot \mathbf{l}_T \odot (\mathbf{1}_T - \mathbf{m}_T)\|_2}{\|\mathbf{w}_T \odot \mathbf{l}_T \odot \mathbf{m}_T\|_2} \\
&\leq \frac{\|\mathbf{w}_T \odot (\mathbf{1}_T - \mathbf{l}_T)\|_2 + \|\mathbf{w}_T \odot \mathbf{l}_T\|_2}{\|\mathbf{w}_T \odot \mathbf{l}_T \odot \mathbf{m}_T\|_2} \\
&= \frac{\|\mathbf{w}_T\|_2}{\|\mathbf{w}_T \odot \mathbf{l}_T \odot \mathbf{m}_T\|_2} \\
&\leq (\rho\sigma)^{-1/2} := a', \quad \text{(B.4)}
\end{aligned}$$

where $C = (1 - 2\rho\sigma) \cdot \lambda \log^{\frac{1}{1+\alpha}}(4/\epsilon) + \sqrt{2 \log(2/\epsilon)}$. We also have

$$\frac{b}{\|\mathbf{w}_T \odot \mathbf{l}_T \odot \mathbf{m}_T\|_2} = ax + c,$$

and hence

$$\mathbb{E}_{X \sim \mathcal{N}(b, \|\mathbf{w}_T \odot \mathbf{l}_T \odot \mathbf{m}_T\|_2^2)}[\text{sign}(X) | \mathbf{y}_H, \mathbf{l}_T, \mathbf{m}_T]$$

$$= \frac{e^{-\frac{(ax+c)^2}{2}}}{\sqrt{2\pi}} \cdot \mathbb{E}_{s \sim Q}\left[ \text{sign}(\|\mathbf{w}_T \odot \mathbf{l}_T \odot \mathbf{m}_T\|_2 \cdot s) \cdot e^{-\frac{s^2}{2} - \log Q(s)} \cdot e^{(ax+c)s} \Big| \mathbf{y}_H, \mathbf{l}_T, \mathbf{m}_T \right]$$

$$= \frac{e^{-\frac{a^2 x^2}{2}}}{\sqrt{2\pi}} \cdot \mathbb{E}_{s \sim Q}\left[ \text{sign}(\|\mathbf{w}_T \odot \mathbf{l}_T \odot \mathbf{m}_T\|_2 \cdot s) \cdot e^{-\frac{s^2}{2} - \log Q(s)} \cdot e^{a(s-c)x + c(s - \frac{1}{2}c)} \Big| \mathbf{y}_H, \mathbf{l}_T, \mathbf{m}_T \right].$$

Then, we have

$$\overline{T_\rho f_\mathbf{x}(\mathbf{z})}$$

$$= \mathbb{E}_{\mathbf{y}_H, \mathbf{l}_T, \mathbf{m}_T}\left[ \mathbb{E}_{X \sim \mathcal{N}(b, \|\mathbf{w}_T \odot \mathbf{l}_T \odot \mathbf{m}_T\|_2^2)}[\text{sign}(X) | \mathbf{y}_H, \mathbf{l}_T, \mathbf{m}_T] \cdot \mathbb{1}[(\mathbf{l}_T, \mathbf{m}_T) \in \mathcal{E}_1] \cdot \mathbb{1}[\mathbf{y}_H \in \mathcal{E}^c] \right]$$

$$= \mathbb{E}_{\mathbf{y}_H, \mathbf{l}_T, \mathbf{m}_T}\Big[ e^{-\frac{a^2 x^2}{2}} \cdot \mathbb{1}[(\mathbf{l}_T, \mathbf{m}_T) \in \mathcal{E}_1] \cdot \mathbb{1}[\mathbf{y}_H \in \mathcal{E}^c]$$

$$\cdot \mathbb{E}_{s \sim Q}\left[ \text{sign}(\|\mathbf{w}_T \odot \mathbf{l}_T \odot \mathbf{m}_T\|_2 \cdot s) \cdot e^{-\frac{s^2}{2} - \log Q(s)} \cdot e^{a(s-c)x + c(s - \frac{1}{2}c)} \Big| \mathbf{y}_H, \mathbf{l}_T, \mathbf{m}_T \right] \Big].$$

We now define a polynomial $\widetilde{p}_\mathbf{z}(\mathbf{x})$ approximating $\overline{T_\rho f_\mathbf{x}(\mathbf{z})}$. To do this, we approximate $e^{-\frac{1}{2}a^2 x^2}$ and $e^{a(s-c)x}$ using polynomials in $x$. First, we use a polynomial $p_1(x)$ to approximate $e^{-\frac{1}{2}a^2 x^2}$. This polynomial is given by the following lemma. We choose the parameters later.

**Lemma B.2** *Let $t \in \mathbb{Z}_+$. Let $x$ be a random variable satisfying the $(\alpha, \lambda)$-strictly sub-exponential tail bound. Then there exists a polynomial $q$ of degree*

$$O\left( (a^2 \lambda^2/2)^{1+1/\alpha} \left( Cb \log(ab\lambda \sqrt{\log(1/\epsilon)}) \right)^{2/\alpha} \left( \log(1/\epsilon) \right)^{\max\{1, 1/2 + 1/\alpha\}} C^{1/\alpha^2} \right)$$

*where $C$ is a sufficiently large constant such that the approximation error $\mathbb{E}_x[(q(x) - e^{-\frac{1}{2}a^2 x^2})^b]$ is upper bounded by $2\epsilon$.*

Second, to approximate $e^{a(s-c)x}$, we use the function $p_2(x, s) = p_k(a(s-c)x) \, \mathbb{1}[|s| \le T]$ where $p_k(x) = 1 + \sum_{i=1}^{k-1} \frac{x^i}{i!}$ is the degree $k-1$ Taylor approximation of $e^x$. We choose degree $k$ and threshold $T$ later. Thus our final approximation of $\overline{T_\rho f_{\mathbf{x}}(\mathbf{z})}$ is

$$
\begin{aligned}
\widetilde{p}_{\mathbf{x}}(\mathbf{y}_H, \mathbf{l}_T, \mathbf{m}_T) &= p_1(x) \cdot \mathbb{E}_{s \sim Q}\big[\operatorname{sign}(\|\mathbf{w}_T \odot \mathbf{l}_T \odot \mathbf{m}_T\|_2 \cdot s) \cdot e^{-\frac{s^2}{2} - \log Q(s)} \\
&\qquad\qquad \cdot e^{c(s - \frac{1}{2}c)} \cdot p_2(x, s)\big|\mathbf{y}_H, \mathbf{l}_T, \mathbf{m}_T\big], \\
\widetilde{p}_{\mathbf{y}_H}(\mathbf{x}) &= \mathbb{E}_{\mathbf{l}_T, \mathbf{m}_T}\big[\widetilde{p}_{\mathbf{x}}(\mathbf{y}_H, \mathbf{l}_T, \mathbf{m}_T) \cdot \mathbb{1}[(\mathbf{l}_T, \mathbf{m}_T) \in \mathcal{E}_1]\big], \\
\widetilde{p}_{\mathbf{z}}(\mathbf{x}) &= \mathbb{E}_{\mathbf{y}_H}\big[\widetilde{p}_{\mathbf{y}_H}(\mathbf{x}) \cdot \mathbb{1}[\mathbf{y}_H \in \mathcal{E}^c]\big].
\end{aligned}
\tag{B.5}
$$

We now want to bound the $L_1$ error term $\mathbb{E}_{\mathbf{x} \sim D_{\mathbf{x}}} \mathbb{E}_{\mathbf{z} \sim \mathcal{N}_\sigma}[|\widetilde{p}_{\mathbf{z}}(\mathbf{x}) - \overline{T_\rho f_{\mathbf{x}}(\mathbf{z})}|]$. To help us analyse the error, we define the "hybrid" function $\bar{p}_{\mathbf{z}}(\mathbf{x})$ such that

$$
\begin{aligned}
\bar{p}_{\mathbf{x}}(\mathbf{y}_H, \mathbf{l}_T, \mathbf{m}_T) &= \frac{e^{-\frac{a^2 x^2}{2}}}{\sqrt{2\pi}} \cdot \mathbb{E}_{s \sim \mathcal{N}(0,1)}\big[\operatorname{sign}(\|\mathbf{w}_T \odot \mathbf{l}_T \odot \mathbf{m}_T\|_2 \cdot s) \cdot e^{-\frac{s^2}{2} - \log Q(s)} \\
&\qquad\qquad \cdot e^{c(s - \frac{1}{2}c)} \cdot p_2(x, s)\big|\mathbf{y}_H, \mathbf{l}_T, \mathbf{m}_T\big], \\
\bar{p}_{\mathbf{z}}(\mathbf{x}) &= \mathbb{E}_{\mathbf{y}_H, \mathbf{l}_T, \mathbf{m}_T}\big[\bar{p}_{\mathbf{x}}(\mathbf{y}_H, \mathbf{l}_T, \mathbf{m}_T) \cdot \mathbb{1}[(\mathbf{l}_T, \mathbf{m}_T) \in \mathcal{E}_1] \cdot \mathbb{1}[\mathbf{y}_H \in \mathcal{E}^c]\big].
\end{aligned}
$$

We have that

$$
\begin{aligned}
&\mathbb{E}_{\mathbf{x} \sim D_{\mathbf{x}}} \mathbb{E}_{\mathbf{z} \sim \mathcal{N}_\sigma}[|\widetilde{p}_{\mathbf{z}}(\mathbf{x}) - \overline{T_{1-\rho} f_{\mathbf{x}}(\mathbf{z})}|] \\
&\le 2 \cdot \mathbb{E}_{\mathbf{x} \sim D_{\mathbf{x}}}\Big[\underbrace{\mathbb{E}_{\mathbf{z} \sim \mathcal{N}_\sigma}[|\bar{p}_{\mathbf{z}}(\mathbf{x}) - \overline{T_{1-\rho} f_{\mathbf{x}}(\mathbf{z})}|]}_{\Delta_3(\mathbf{x})} + \underbrace{\mathbb{E}_{\mathbf{z} \sim \mathcal{N}_\sigma}[|\widetilde{p}_{\mathbf{z}}(\mathbf{x}) - \bar{p}_{\mathbf{z}}(\mathbf{x})|]}_{\Delta_4(\mathbf{x})}\Big].
\end{aligned}
$$

We now bound $\Delta_3(\mathbf{x})$ and $\Delta_4(\mathbf{x})$ separately. We have that

$$
\begin{aligned}
\Delta_3(\mathbf{x}) \le \mathbb{E}_{\mathbf{z}}\Big[&\mathbb{E}_{\mathbf{y}_H, \mathbf{l}_T, \mathbf{m}_T}\big[e^{-\frac{1}{2}a^2 x^2} \cdot \mathbb{E}_{s \sim \mathcal{N}(0,1)}\big[e^{-\frac{s^2}{2} - \log Q(s)} \cdot e^{c(s - \frac{1}{2}c)} \cdot |e^{a(s-c)x} - p_2(x, s)|\big] \\
&\cdot \mathbb{1}[(\mathbf{l}_T, \mathbf{m}_T) \in \mathcal{E}_1] \cdot \mathbb{1}[\mathbf{y}_H \in \mathcal{E}^c]\big]\Big].
\end{aligned}
$$

Observe that $\Delta_3(\mathbf{x})$ can be bounded as the expected sum of the following two terms:

$$
\begin{aligned}
&\Delta_{31}(\mathbf{x}, \mathbf{y}_H, \mathbf{l}_T, \mathbf{m}_T) \\
&= \frac{e^{-\frac{a^2 x^2}{2}}}{\sqrt{2\pi}} \cdot \mathbb{E}_{s \sim Q}\Big[e^{-\frac{s^2}{2} - \log Q(s)} \cdot e^{c(s - \frac{1}{2}c)} \cdot \frac{e^{|a(s-c)x|}}{k!} \cdot |a(s-c)x|^k \cdot \mathbb{1}[|s| \le T]\Big], \\
&\Delta_{32}(\mathbf{x}, \mathbf{y}_H, \mathbf{l}_T, \mathbf{m}_T) \\
&= \frac{e^{-\frac{a^2 x^2}{2}}}{\sqrt{2\pi}} \cdot \mathbb{E}_{s \sim Q}\Big[e^{-\frac{s^2}{2} - \log Q(s)} \cdot e^{c(s - \frac{1}{2}c)} \cdot e^{a(s-c)x} \cdot \mathbb{1}[|s| > T]\Big],
\end{aligned}
$$

where the first term's bound comes from the fact that $|p_k(x) - e^x| \le \frac{e^{|x|}}{k!} \cdot |x|^k$.

We first bound $\Delta_{31}$. We have that

$$
\Delta_{31}(\mathbf{x}, \mathbf{y}_H, \mathbf{l}_T, \mathbf{m}_T)
$$

$$
\leq \frac{e^{-\frac{a^2 x^2}{2}}}{\sqrt{2\pi}} \cdot \mathbb{E}_{s \sim Q}\left[e^{-\frac{s^2}{2} - \log Q(s)} \cdot e^{c(s - \frac{1}{2}c)} \cdot e^{|a(s-c)x|} \cdot \mathbb{1}[|s| \leq T]\right] \cdot \frac{(|acx| + |aTx|)^k}{k!}
$$

$$
\leq \frac{(|acx| + |aTx|)^k}{k!} \cdot \mathbb{E}_{s \sim Q}\left[\frac{e^{-\frac{1}{2}(s - (ax+c))^2}}{\sqrt{2\pi} \cdot Q(s)} \cdot \mathbb{1}[|s| \leq T]\right]
$$

$$
+ \frac{(|acx| + |aTx|)^k}{k!} \cdot \mathbb{E}_{s \sim Q}\left[\frac{e^{-\frac{1}{2}(s - (c-ax))^2}}{\sqrt{2\pi} \cdot Q(s)} \cdot \mathbb{1}[|s| \leq T]\right]
$$

$$
\leq \frac{(|acx| + |aTx|)^k}{(k!)^2} \cdot \mathbb{E}_{s \sim Q}\left[\frac{\mathcal{N}(s; ax + c, 1)}{Q(s)}\right]
$$

$$
+ \frac{(|acx| + |aTx|)^{2k}}{(k!)^2} \cdot \mathbb{E}_{s \sim Q}\left[\frac{\mathcal{N}(s; c - ax, 1)}{Q(s)}\right]
$$

$$
= \frac{2(|acx| + |aTx|)^k}{k!}
$$

where the second inequality is by $e^{|a(s-c)x|} \leq e^{a(s-c)x} + e^{-a(s-c)x}$.

Then, we have

$$
\mathbb{E}_{\mathbf{x} \sim \mathcal{D}_{\mathbf{x}}}\left[\mathbb{E}_{\mathbf{z} \sim \mathcal{N}_\sigma}\left[\mathbb{E}_{\mathbf{y}_H, \mathbf{l}_T, \mathbf{m}_T}\left[\Delta_{31}(\mathbf{x}, \mathbf{y}_H, \mathbf{l}_T, \mathbf{m}_T) \cdot \mathbb{1}[(\mathbf{l}_T, \mathbf{m}_T) \in \mathcal{E}_1] \cdot \mathbb{1}[\mathbf{y}_H \in \mathcal{E}^c]\right]\right]\right]
$$

$$
\leq 2\mathbb{E}_{\mathbf{x} \sim \mathcal{D}_{\mathbf{x}}}\left[\mathbb{E}_{\mathbf{z} \sim \mathcal{N}_\sigma}\left[\mathbb{E}_{\mathbf{y}_H, \mathbf{l}_T, \mathbf{m}_T}\left[\frac{(|acx| + |aTx|)^k}{k!} \cdot \mathbb{1}[(\mathbf{l}_T, \mathbf{m}_T) \in \mathcal{E}_1] \cdot \mathbb{1}[\mathbf{y}_H \in \mathcal{E}^c]\right]\right]\right]
$$

$$
= 2\mathbb{E}_{\mathbf{z} \sim \mathcal{N}_\sigma}\left[\mathbb{E}_{\mathbf{y}_H, \mathbf{m}_T}\left[\mathbb{E}_{\mathbf{x} \sim \mathcal{D}_{\mathbf{x}}}\left[\frac{(|acx| + |aTx|)^k}{k!} \cdot \mathbb{1}[(\mathbf{l}_T, \mathbf{m}_T) \in \mathcal{E}_1] \cdot \mathbb{1}[\mathbf{y}_H \in \mathcal{E}^c]\right]\right]\right].
$$

Under condition $(\mathbf{l}_T, \mathbf{m}_T) \in \mathcal{E}_1$ and condition $\mathbf{y}_H \in \mathcal{E}^c$, we have (B.3) and (B.4) and hence

$$
\mathbb{E}_{\mathbf{x} \sim \mathcal{D}_{\mathbf{x}}}\left[\frac{(|acx| + |aTx|)^k}{k!} \cdot \mathbb{1}[(\mathbf{l}_T, \mathbf{m}_T) \in \mathcal{E}_1] \cdot \mathbb{1}[\mathbf{y}_H \in \mathcal{E}^c]\right]
$$

$$
\leq \frac{|a'|^k (|c'| + |T|)^k}{k!} \cdot \mathbb{E}_{\mathbf{x} \sim D}[|x|^k]
$$

$$
\leq \frac{|a'|^k (|c'| + |T|)^k}{k!} \cdot C\lambda^k \cdot \left(\frac{k}{1+\alpha}\right)^{\frac{k}{1+\alpha} + \frac{1}{2}} e^{-\frac{k}{1+\alpha} + \frac{1+\alpha}{12k}}
$$

$$
\leq \epsilon,
$$

when

$$
k \geq \left(C'|a'|(|c'| + |T|)\lambda \log(1/\epsilon)\right)^{1 + \frac{1}{\alpha}} + (1 + \alpha) \tag{B.6}
$$

and $C'$ is a large enough constant. The second inequality used Lemma C.5.

We now bound $\Delta_{32}$. We have that

$$
\Delta_{32}(\mathbf{x}, \mathbf{y}_H, \mathbf{l}_T, \mathbf{m}_T) = \mathbb{E}_{s \sim Q}\left[\frac{e^{-\frac{1}{2}(s - (ax+c))^2}}{Q(s)} \cdot \mathbb{1}[|s| > T]\right]
$$

$$
\leq \sqrt{\mathbb{E}_{s \sim Q}\left[\left(\frac{\mathcal{N}(s; ax + c, 1)}{Q(s)}\right)^2\right] \cdot \mathbb{P}_{s \sim Q}[|s| > T]}
$$

$$
\leq \sqrt{Ce^{|ax+c|} \cdot e^{-T}}
$$

$$
\leq C'e^{\frac{1}{2}(|ax| + |c|)} \cdot e^{-T/2}.
$$

The third inequality is based on the following claim.

**Lemma B.3** *Define the distribution $Q$ on $\mathbb{R}$ with density function $Q(s) = e^{-|s|}/2$. Then there exist a universal constant $C$ such that for every $b \in \mathbb{R}$, it holds that*

$$\mathbb{E}_{s \sim Q}\left[\left(\frac{\mathcal{N}(s; b, 1)}{Q(s)}\right)^2\right] \leq C e^{|b|}.$$

Thus, we have that

$$\mathbb{E}_{\mathbf{x} \sim \mathcal{D}_{\mathbf{x}}}\left[\mathbb{E}_{\mathbf{z} \sim \mathcal{N}_\sigma}\left[\mathbb{E}_{\mathbf{y}_H, \mathbf{l}_T, \mathbf{m}_T}\left[\Delta_{32}(\mathbf{x}, \mathbf{y}_H, \mathbf{l}_T, \mathbf{m}_T) \cdot \mathbb{1}[(\mathbf{l}_T, \mathbf{m}_T) \in \mathcal{E}_1] \cdot \mathbb{1}[\mathbf{y}_H \in \mathcal{E}^c]]\right]\right]$$

$$\leq \mathbb{E}_{\mathbf{x} \sim \mathcal{D}_{\mathbf{x}}}\left[\mathbb{E}_{\mathbf{z} \sim \mathcal{N}_\sigma}\left[\mathbb{E}_{\mathbf{y}_H, \mathbf{l}_T, \mathbf{m}_T}\left[C' e^{\frac{1}{2}(|ax| + |c|)} \cdot e^{-T/2} \cdot \mathbb{1}[(\mathbf{l}_T, \mathbf{m}_T) \in \mathcal{E}_1] \cdot \mathbb{1}[\mathbf{y}_H \in \mathcal{E}^c]]\right]\right]$$

$$= \mathbb{E}_{\mathbf{z} \sim \mathcal{N}_\sigma}\left[\mathbb{E}_{\mathbf{y}_H, \mathbf{l}_T, \mathbf{m}_T}\left[\mathbb{E}_{\mathbf{x} \sim \mathcal{D}_{\mathbf{x}}}\left[C' e^{\frac{1}{2}(|ax| + |c|)} \cdot e^{-T/2} \cdot \mathbb{1}[(\mathbf{l}_T, \mathbf{m}_T) \in \mathcal{E}_1] \cdot \mathbb{1}[\mathbf{y}_H \in \mathcal{E}^c]]\right]\right]$$

$$\leq \mathbb{E}_{\mathbf{z} \sim \mathcal{N}_\sigma}\left[\mathbb{E}_{\mathbf{y}_H, \mathbf{l}_T, \mathbf{m}_T}\left[\mathbb{E}_{\mathbf{x} \sim \mathcal{D}_{\mathbf{x}}}\left[C' e^{\frac{1}{2}|a'x|} \cdot \mathbb{1}[(\mathbf{l}_T, \mathbf{m}_T) \in \mathcal{E}_1] \cdot \mathbb{1}[\mathbf{y}_H \in \mathcal{E}^c] \cdot e^{\frac{|c'| - T}{2}}\right]\right]\right]$$

$$\leq \mathbb{E}_{\mathbf{z} \sim \mathcal{N}_\sigma}\left[\mathbb{E}_{\mathbf{y}_H, \mathbf{l}_T, \mathbf{m}_T}\left[\mathbb{E}_{\mathbf{x} \sim \mathcal{D}_{\mathbf{x}}}\left[C'' e^{(a'\lambda)^{1 + \frac{1}{\alpha}}/2} \cdot e^{\frac{|c'| - T}{2}}\right]\right]\right]$$

$$= C'' e^{(a'\lambda)^{1 + \frac{1}{\alpha}}/2} \cdot e^{\frac{|c'| - T}{2}}$$

$$\leq \epsilon,$$

when

$$T \geq (a'\lambda)^{1 + \frac{1}{\alpha}} + |c'| + 2\log(C''/\epsilon).$$

By (B.3) and (B.4), we can take

$$T = O\left((\rho\sigma)^{-\frac{1}{2}(1 + \frac{1}{\alpha})}\lambda^{1 + \frac{1}{\alpha}} \log(1/\epsilon)\right). \tag{B.7}$$

The third last inequality is by Lemma C.6. Plugging this into the bound for $k$ in (B.6), we can take

$$k \leq \left(C' a'\lambda\left(2c' + (a'\lambda)^{1 + \frac{1}{\alpha}} + 2\log(C''/\epsilon)\right)\log(1/\epsilon)\right)^{1 + \frac{1}{\alpha}} + (1 + \alpha)$$

for constant $C', C''$. By (B.3) and (B.4), we know that

$$k \leq \left(C''' \lambda^{2 + \frac{1}{\alpha}}(\rho\sigma)^{-(1 + \frac{1}{2\alpha})}\log^3(1/\epsilon)\right)^{1 + \frac{1}{\alpha}} + (1 + \alpha), \tag{B.8}$$

where $C'''$ is a large constant.

We now bound $\Delta_4(\mathbf{x})$. We have that

$$\Delta_4(\mathbf{x})$$

$$\leq \mathbb{E}_{\mathbf{z}}\left[\mathbb{E}_{\mathbf{y}_H, \mathbf{l}_T, \mathbf{m}_T}\left[|\widetilde{p}_{\mathbf{x}}(\mathbf{y}_H, \mathbf{l}_H, \mathbf{m}_T) - \bar{p}_{\mathbf{x}}(\mathbf{y}_H, \mathbf{l}_H, \mathbf{m}_T)| \cdot \mathbb{1}[(\mathbf{l}_T, \mathbf{m}_T) \in \mathcal{E}_1] \cdot \mathbb{1}[\mathbf{y}_H \in \mathcal{E}^c]]\right]\right]$$

$$= \frac{1}{\sqrt{2\pi}}\mathbb{E}_{\mathbf{z}}\left[\mathbb{E}_{\mathbf{y}_H, \mathbf{l}_T, \mathbf{m}_T}\left[|e^{-\frac{1}{2}a^2 x^2} - p_1(x)| \cdot \left|\mathbb{E}_{s \sim Q}[e^{-\frac{s^2}{2} - \log Q(s) + c(s - \frac{c}{2})} \cdot p_2(x, s)]\right|\right.\right.$$

$$\left.\left. \cdot \mathbb{1}[(\mathbf{l}_T, \mathbf{m}_T) \in \mathcal{E}_1] \cdot \mathbb{1}[\mathbf{y}_H \in \mathcal{E}^c]]\right]\right]$$

Notice that

$$\left(\mathbb{E}_{s \sim Q}[e^{-\frac{s^2}{2} - \log Q(s) + c(s - \frac{c}{2})} \cdot p_2(x, s)]\right)^2$$

$$\leq \mathbb{E}_{s \sim Q}[e^{-s^2 - 2\log Q(s) + c(2s - c)}] \cdot \mathbb{E}_{s \sim Q}[p_2(x, s)^2]$$

$$\leq \mathbb{E}_{s \sim Q}\left[\left(\frac{\mathcal{N}(s; c, 1)}{Q(s)}\right)^2\right] \cdot \mathbb{E}_{s \sim Q}\left[\left(\sum_{i=0}^{k-1} \frac{(a(s - c)x)^i}{i!}\right)^2 \cdot \mathbb{1}[|s| \leq T]\right]$$

$$\leq C e^{|c|} \cdot \left(\sum_{i=0}^{k-1} \frac{|a|^i(|T| + |c|)^i|x|^i}{i!}\right)^2,$$

where the last inequality is by Lemma B.3. Thus, we have

$$\mathbb{E}_{\mathbf{x}\sim\mathcal{D}_{\mathbf{x}}}[\Delta_4(\mathbf{x})]$$

$$\leq \mathbb{E}_{\mathbf{z}}\Big[\mathbb{E}_{\mathbf{y}_H,\mathbf{l}_T,\mathbf{m}_T}\Big[\mathbb{E}_{\mathbf{x}\sim\mathcal{D}_{\mathbf{x}}}\big[\big|e^{-\frac{1}{2}a^2x^2}-p_1(x)\big|\cdot\big|\mathbb{E}_{s\sim Q}[e^{-\frac{s^2}{2}-\log Q(s)+c(s-\frac{c}{2})}\cdot p_2(x,s)]\big|\big]$$

$$\cdot\mathbb{1}[(\mathbf{l}_T,\mathbf{m}_T)\in\mathcal{E}_1]\cdot\mathbb{1}[\mathbf{y}_H\in\mathcal{E}^c]\big]\Big]$$

$$\leq \mathbb{E}_{\mathbf{z}}\Big[\mathbb{E}_{\mathbf{y}_H,\mathbf{l}_T,\mathbf{m}_T}\Big[\mathbb{E}_{\mathbf{x}\sim\mathcal{D}_{\mathbf{x}}}\big[\big|e^{-\frac{1}{2}a^2x^2}-p_1(x)\big|\cdot C^{\frac{1}{2}}e^{\frac{|c|}{2}}\cdot\Big(\sum_{i=0}^{k-1}\frac{|a|^i(|T|+|c|)^i|x|^i}{i!}\Big)$$

$$\cdot\mathbb{1}[(\mathbf{l}_T,\mathbf{m}_T)\in\mathcal{E}_1]\cdot\mathbb{1}[\mathbf{y}_H\in\mathcal{E}^c]\big]\Big]\Big]$$

Denote

$$\Delta_5(\mathbf{y}_H,\mathbf{l}_T,\mathbf{m}_T):=\mathbb{E}_{\mathbf{x}\sim\mathcal{D}_{\mathbf{x}}}\Big[\big|e^{-\frac{1}{2}a^2x^2}-p_1(x)\big|\cdot C^{\frac{1}{2}}e^{\frac{|c|}{2}}\cdot\Big(\sum_{i=0}^{k-1}\frac{|a|^i(|T|+|c|)^i|x|^i}{i!}\Big)$$

$$\cdot\mathbb{1}[(\mathbf{l}_T,\mathbf{m}_T)\in\mathcal{E}_1]\cdot\mathbb{1}[\mathbf{y}_H\in\mathcal{E}^c]\Big].$$

We have

$$\Delta_5(\mathbf{y}_H,\mathbf{l}_T,\mathbf{m}_T)$$

$$\leq C'e^{|c'|/2}\sqrt{\mathbb{E}_{\mathbf{x}\sim\mathcal{D}_{\mathbf{x}}}\big[\big(e^{-\frac{1}{2}a^2x^2}-p_1(x)\big)^2\cdot\mathbb{1}[(\mathbf{l}_T,\mathbf{m}_T)\in\mathcal{E}_1]\cdot\mathbb{1}[\mathbf{y}_H\in\mathcal{E}^c]\big]}$$

$$\cdot\sqrt{\mathbb{E}_{\mathbf{x}\sim\mathcal{D}_{\mathbf{x}}}\Big[\Big(\sum_{i=0}^{k-1}\frac{|a|^i(|T|+|c|)^i|x|^i}{i!}\Big)^2\cdot\mathbb{1}[(\mathbf{l}_T,\mathbf{m}_T)\in\mathcal{E}_1]\cdot\mathbb{1}[\mathbf{y}_H\in\mathcal{E}^c]\Big]}$$

$$\leq C'e^{|c'|/2}\cdot\delta\cdot\sqrt{\mathbb{E}_{\mathbf{x}\sim\mathcal{D}_{\mathbf{x}}}\Big[\Big(\sum_{i=0}^{k-1}\frac{|a|^i(|T|+|c|)^i|x|^i}{i!}\Big)^2\cdot\mathbb{1}[(\mathbf{l}_T,\mathbf{m}_T)\in\mathcal{E}_1]\cdot\mathbb{1}[\mathbf{y}_H\in\mathcal{E}^c]\Big]}$$

$$\leq C'e^{|c'|/2}\cdot\delta\cdot\sqrt{\Big(\sum_{i=0}^{k-1}\frac{|a'|^i(|T|+|c'|)^i}{i!}\Big)^2\cdot\max_{1\leq i\leq k-1}C\lambda^{2i}\cdot\Big(\frac{2i}{1+\alpha}\Big)^{\frac{2i}{1+\alpha}+\frac{1}{2}}e^{-\frac{2i}{1+\alpha}+\frac{1+\alpha}{24i}}}$$

$$\leq C''\delta e^{|c'|/2+|a'|(|T|+|c'|)}\lambda^k\Big(\frac{2k}{1+\alpha}\Big)^{\frac{k}{1+\alpha}+\frac{1}{4}}e^{\frac{1+\alpha}{48}}$$

$$\leq \epsilon,$$

when $\delta$ is chosen accordingly. The third inequality is by Lemma C.5.

By Lemma B.2 and taking the exponent $b$ as 2, the degree of $p_1(x)$ required to get the error is

$$O\Big((a^2\lambda^2/2)^{1+1/\alpha}\big(C\log(a\lambda\sqrt{\log(1/\delta)})\big)^{2/\alpha}\big(\log(1/\delta)\big)^{\max\{1,\frac{1}{2}+\frac{1}{\alpha}\}}C^{1/\alpha^2}\Big)$$

$$= O\Big((a'^2\lambda^2/2)^{1+1/\alpha}\big(C\log(a'\lambda)\big)^{2/\alpha}$$

$$\cdot\big(\log(1/\epsilon)+|a'|(|T|+|c'|)+(k+1/4)\log(2k\lambda)\big)^{2/\alpha+1}C^{1/\alpha^2}\Big)$$

$$= O\Big((a'^2\lambda^2/2)^{1+1/\alpha}\big(C\log(a'\lambda)\big)^{2/\alpha}\big(\log(1/\epsilon)+|a'|(|T|+|c'|)+k^{3/2}\lambda^{1/2}\big)^{2/\alpha+1}C^{1/\alpha^2}\Big).$$

By using (B.3) and (B.4) and plugging in the order of $k, T$ in (B.7) and (B.8), we know the degree of $p_1(x)$ is

$$O\Big(((\rho\sigma)^{-1}\lambda^2/2)^{1+\frac{1}{\alpha}}\big(C\log((\rho\sigma)^{-\frac{1}{2}}\lambda)\big)^{\frac{2}{\alpha}}$$

$$\cdot\big(C\lambda^{\frac{3}{2}(2+\frac{1}{\alpha})(1+\frac{1}{\alpha})+\frac{1}{2}}(\rho\sigma)^{-\frac{3}{2}(1+\frac{1}{2\alpha})(1+\frac{1}{\alpha})}\log^{\frac{9}{2}(1+\frac{1}{\alpha})}(1/\epsilon)\big)^{1+\frac{2}{\alpha}}C^{\frac{1}{\alpha^2}}\Big)$$

$$= O\Big(\big(C(\rho\sigma)^{-\frac{1}{2}}\lambda\log(1/\epsilon)\big)^{6(1+\frac{1}{\alpha})^3}\Big).$$

Putting everything together, we get that the degree of $p_{\mathbf{z}}(\mathbf{x})$ is at most $2H + \deg(p_1) + \deg(p_2)$ which is

$$O\big(\log(1+\lambda)/\alpha^2 + \log(1/\epsilon)\log(1/\alpha)/\rho\sigma\alpha^2\big) + O\Big(\big(C(\rho\sigma)^{-\frac{1}{2}}\lambda\log(1/\epsilon)\big)^{6(1+\frac{1}{\alpha})^3}\Big).$$

Plugging in the order of $\rho = \Omega(\epsilon^2)$ and $\alpha = \Omega(\rho\sigma/\sqrt{\log(1/\epsilon)})$, the degree can be bounded by

$$O\Big(\big(C\sigma^{-\frac{1}{2}}\lambda\log(1/\epsilon)/\epsilon\big)^{6(1+\frac{1}{\alpha})^3}\Big).$$

## C   Proofs of Auxiliary Lemmas

### C.1   Proof of Theorem 4.2

**Proof**  Let $f^*$ be the optimal halfspace that achieves $\mathrm{opt}_\sigma$. Let $p_{\mathbf{z}}$ be the polynomial of degree at most $d$ such that

$$\mathbb{E}_{\mathbf{z}\sim\mathcal{D}_\sigma,\mathbf{x}\sim\mathcal{D}_{\mathbf{x}}}[|p_{\mathbf{z}}(\mathbf{x}) - f^*(\mathbf{x}\odot\mathbf{z})|] \le \epsilon. \tag{C.1}$$

Consider the sample dataset $S = \{(\mathbf{x}_i, y_i)\}_{i=1}^N$ in a single run of Algorithm 1. Let $p_S$ be the polynomial chosen by the algorithm and let $h_S$ be the corresponding hypothesis that the algorithm outputs. By the proof of Theorem 4.1 in Kalai et al. (2008), we have that

$$\frac{1}{N}\sum_{i=1}^N \mathbb{1}[h_S(\mathbf{x}_i) \ne y_i] \le \frac{1}{2N}\sum_{i=1}^N |y_i - p_S(\mathbf{x}_i)|.$$

Notice that $p_S$ is the minimizer of the error, and thus beats any polynomial $p_{\mathbf{z}}$ we choose, we have

$$\frac{1}{2N}\sum_{i=1}^N |y_i - p_S(\mathbf{x}_i)| \le \mathbb{E}_{\mathbf{z}\sim\mathcal{N}_\sigma}\left[\frac{1}{2N}\sum_{i=1}^N |y_i - p_{\mathbf{z}}(\mathbf{x}_i)|\right]$$

$$\le \underbrace{\mathbb{E}_{\mathbf{z}\sim\mathcal{N}_\sigma}\left[\frac{1}{2N}\sum_{i=1}^N |f^*(\mathbf{x}_i\odot\mathbf{z}) - p_{\mathbf{z}}(\mathbf{x}_i)|\right]}_{\Delta_1(S)} + \underbrace{\mathbb{E}_{\mathbf{z}\sim\mathcal{N}_\sigma}\left[\frac{1}{2N}\sum_{i=1}^N |y_i - f^*(\mathbf{x}_i\odot\mathbf{z})|\right]}_{\Delta_2(S)}.$$

By (C.1), we can bound $\Delta_1(S)$ as follows:

$$\mathbb{E}_{S\sim\mathcal{D}^{\otimes n}}[\Delta_1(S)] = \frac{1}{2}\mathbb{E}_{\mathbf{x}\sim\mathcal{D}_{\mathbf{x}},\mathbf{z}\sim\mathcal{D}_\sigma}[|f^*(\mathbf{x}\odot\mathbf{z}) - p_{\mathbf{z}}(\mathbf{x})|] \le \frac{1}{2}\epsilon.$$

By the optimality of $f^*$, we have

$$\mathbb{E}_{S\sim\mathcal{D}^{\otimes n}}[\Delta_1(S)] = \frac{1}{2}\mathbb{E}_{(\mathbf{x},y)\sim\mathcal{D},\mathbf{z}\sim\mathcal{D}_\sigma}[|y - f^*(\mathbf{x}\odot\mathbf{z})|]$$

$$= \mathbb{E}_{(\mathbf{x},y)\sim\mathcal{D},\mathbf{z}\sim\mathcal{D}_\sigma}[\mathbb{1}[y \ne f^*(\mathbf{x}\odot\mathbf{z})]] = \mathrm{opt}_\sigma.$$

Thus, we obtain

$$\mathbb{E}_{S\sim\mathcal{D}^{\otimes n}}\left[\frac{1}{N}\sum_{i=1}^N \mathbb{1}[h_S(\mathbf{x}_i) \ne y_i]\right] \le \mathrm{opt}_\sigma + \frac{1}{2}\epsilon.$$

Since our hypothesis $h_S$ is a polynomial threshold function of degree $d$ on $n$ variables, VC theory tells us that for $N = \mathrm{poly}(n^d/\epsilon)$, we have that

$$\mathbb{E}_{S\sim\mathcal{D}^{\otimes n}}\left[\mathbb{P}_{(\mathbf{x},y)\sim\mathcal{D}}[h_S(\mathbf{x}) \ne y]\right] \le \mathrm{opt}_\sigma + \frac{3}{4}\epsilon.$$

By Markov's inequality, on any single repetition of the algorithm, we have that

$$\mathbb{P}_{S\sim\mathcal{D}^{\otimes n}}\left[\mathbb{P}_{(\mathbf{x},y)\sim\mathcal{D}}[h_S(\mathbf{x}) \ne y] \ge \mathrm{opt}_\sigma + \frac{7}{8}\epsilon\right] \le \frac{\mathrm{opt}_\sigma + \frac{3}{4}\epsilon}{\mathrm{opt}_\sigma + \frac{7}{8}} \le 1 - \frac{\epsilon}{16}.$$

Hence, after $r = O(\log(1/\delta)/\epsilon)$ repetitions of the algorithm, with probability at least $1 - \delta/2$, one of them will have $\mathbb{P}_{(\mathbf{x},y)\sim\mathcal{D}}[h_S(\mathbf{x}) \ne y] \le \mathrm{opt}_\sigma + \frac{7}{8}\epsilon$. In this case, using an independent set of size $O(\log(1/\delta)/\epsilon^2)$, we probability at most $\delta/2$, we will choose one with error $> \mathrm{opt}_\sigma + \epsilon$. ∎

## C.2 Proof of Lemma B.3

**Proof** The proof below is straightforward calculation by completing the squares.

$$
\begin{aligned}
\mathbb{E}_{s \sim Q}\left[\left(\frac{\mathcal{N}(s; b, 1)}{Q(s)}\right)^2\right] &= \frac{1}{2} \int_{\mathbb{R}} \frac{\frac{1}{2\pi} e^{-(s-b)^2}}{\frac{1}{4} e^{-2|s|}} \cdot e^{-|s|} ds \\
&= \frac{1}{\pi} \int_{\mathbb{R}} e^{-(s-b)^2 + |s|} ds \\
&\leq \frac{1}{\pi} \int_{\mathbb{R}} e^{-(s-b)^2 + s} ds + \frac{1}{\pi} \int_{\mathbb{R}} e^{-(s-b)^2 - s} ds \\
&= \frac{1}{\pi} \int_{\mathbb{R}} e^{-(s-b+\frac{1}{2})^2 + b + \frac{1}{4}} ds + \frac{1}{\pi} \int_{\mathbb{R}} e^{-(s-b-\frac{1}{2})^2 - b + \frac{1}{4}} ds \\
&= \frac{1}{\sqrt{\pi}} e^{b + \frac{1}{4}} + \frac{1}{\sqrt{\pi}} e^{-b + \frac{1}{4}} \\
&\leq \frac{2e^{\frac{1}{4}}}{\sqrt{\pi}} e^{|b|}
\end{aligned}
$$

∎

## C.3 Proof of Lemma B.2

**Proof** Let $p(x) = \sum_{i=0}^{\deg(p)} c_i x^i$ be the polynomial obtained from Lemma C.4 with error $\epsilon/2$ and $T = \omega(\log(1/\epsilon))$ to be choosen later. Our final polynomial is $q(x) = p(\frac{1}{2} a^2 x^2)$. Clearly, $\deg(q) = 2 \cdot \deg(p) = O(\sqrt{T \log(1/\epsilon)})$. We now bound the error.

$$
\begin{aligned}
&\mathbb{E}_x[(q(x) - e^{-\frac{1}{2} a^2 x^2})^b] \\
&= \mathbb{E}_x\left[(q(x) - e^{-\frac{1}{2} a^2 x^2})^b \cdot \mathbb{1}[a^2 x^2/2 < T]\right] + \mathbb{E}_x\left[(q(x) - e^{-\frac{1}{2} a^2 x^2})^b \cdot \mathbb{1}[a^2 x^2/2 \geq T]\right] \\
&= \epsilon + \sqrt{\mathbb{E}_x\left[(q(x) - e^{-\frac{1}{2} a^2 x^2})^{2b}\right] \cdot \mathbb{E}_x\left[\mathbb{1}[a^2 x^2/2 \geq T]\right]} \\
&\leq \epsilon + \sqrt{\mathbb{E}_x\left[(q(x) - e^{-\frac{1}{2} a^2 x^2})^{2b}\right] \cdot 2e^{-\left(\frac{2T}{a^2 \lambda^2}\right)^{\frac{1+\alpha}{2}}}},
\end{aligned}
$$

where the last inequality is by the tail bound of $(\alpha, \lambda)$-strictly sub-exponential random variable.

We now bound $\mathbb{E}_x\left[(q(x) - e^{-\frac{1}{2} a^2 x^2})^{2b}\right]$. We have that

$$
\begin{aligned}
\mathbb{E}_x\left[(q(x) - e^{-\frac{1}{2} a^2 x^2})^{2b}\right] &\leq \mathbb{E}_x\left[(|q(x)| + 1)^{2b}\right] \\
&\leq \mathbb{E}_x\left[\left(1 + \sum_{i=0}^{\deg(p)} \frac{|c_i|}{2^i} a^{2i} x^{2i}\right)^{2b}\right] \\
&\leq (\deg(p) + 2)^{2b} \cdot \max_{i=0}^{\deg(p)} |c_i|^{2b} \cdot \max_{i=0}^{\deg(p)} \frac{a^{4bi} \mathbb{E}_x[x^{4bi}]}{2^{2bi}} \\
&\leq e^{Cb\sqrt{T \log(1/\epsilon)}} \cdot \max_{i=0}^{\deg(p)} \frac{a^{4bi} \mathbb{E}_x[x^{4bi}]}{2^{2bi}} \\
&\leq e^{Cb\sqrt{T \log(1/\epsilon)}} \cdot \max_{i=1}^{\deg(p)} \frac{a^{4bi}}{2^{2bi}} \cdot \frac{C' bi \lambda^{4bi}}{1+\alpha} \cdot \left(\frac{4bi}{1+\alpha}\right)^{\frac{4bi}{1+\alpha} - \frac{1}{2}} e^{-\frac{4bi}{1+\alpha} + \frac{1+\alpha}{48bi}} \\
&\leq e^{Cb\sqrt{T \log(1/\epsilon)} + C' b\sqrt{T \log(1/\epsilon)} \cdot \log(ab\lambda\sqrt{T \log(1/\epsilon)}) + \frac{1+\alpha}{48b}} \\
&\leq e^{C'' b(1+\alpha)\sqrt{T \log(1/\epsilon)} \cdot \log(ab\lambda\sqrt{T \log(1/\epsilon)})},
\end{aligned}
$$

where the third last inequality is by Lemma C.5. Here $C, C', C''$ are large enough constant. Putting it all together, we get that

$$
\mathbb{E}_x\left[(q(x) - e^{-\frac{1}{2} a^2 x^2})^{2b}\right] \cdot 2e^{-\left(\frac{2T}{a^2 \lambda^2}\right)^{\frac{1+\alpha}{2}}} \leq 2e^{C'' b\sqrt{T \log(1/\epsilon)} \cdot \log(ab\lambda\sqrt{T \log(1/\epsilon)}) - \left(\frac{2T}{a^2 \lambda^2}\right)^{\frac{1+\alpha}{2}}}.
$$

Choosing

$$T = \Omega\Big( (a^2\lambda^2/2)^{2(1+1/\alpha)} \big( C''' b^2 \log^2(ab\lambda\sqrt{\log(1/\epsilon)}) \big)^{2/\alpha} \big( \log(1/\epsilon) \big)^{\max\{1,2/\alpha\}} (C''')^{1/\alpha^2} \Big)$$

where $C'''$ is a large constant makes the total error less than $2\epsilon$. Since $T$ is $\omega(\log(1/\epsilon))$, the degree of the final polynomial is $O(\sqrt{T\log(1/\epsilon)})$ which is

$$T = O\Big( (a^2\lambda^2/2)^{1+1/\alpha} \big( C''' b^2 \log^2(ab\lambda\sqrt{\log(1/\epsilon)}) \big)^{1/\alpha} \big( \log(1/\epsilon) \big)^{\max\{1,1/2+1/\alpha\}} (C''')^{1/2\alpha^2} \Big).$$

∎

## C.4 Other Auxiliary Lemmas

**Lemma C.1** *Suppose $\mathbf{x}$ is a distribution on $\{\pm 1\}^n$ that is $(\alpha,\lambda)$-strictly subexponential and $\mathbf{y}$ distributed as $\mathcal{N}_{1-\rho}(\mathbf{z})$. For any fixed $T \subseteq [n]$ and fixed $\mathbf{z}$, with probability at least $1-\epsilon$ of $\mathbf{x}$, it holds that*

$$\mathbb{P}_{\mathbf{y}}\Big[ |\langle \mathbf{u}_T, \mathbf{y}_T \rangle| \leq C \cdot \|\mathbf{u}_T\|_2 \Big] \geq 1-\epsilon,$$

*where $\mathbf{u} = \mathbf{w} \odot \mathbf{x}$ and $C = (1-2\rho\sigma) \cdot \lambda \log^{\frac{1}{1+\alpha}}(4/\epsilon) + \sqrt{2\log(2/\epsilon)}$.*

**Proof** By Hoeffding's inequality, we have

$$\mathbb{P}_{\mathbf{y}}\Big[ \big| \langle \mathbf{u}_T, \mathbf{y}_T \rangle - \mathbb{E}_{\mathbf{y}}[\langle \mathbf{u}_T, \mathbf{y}_T \rangle] \big| \geq t \Big] \leq 2\exp\Big( -\frac{t^2}{2\|\mathbf{u}_T\|_2^2} \Big).$$

Therefore, with probability at least $1-\epsilon$ it holds that

$$\langle \mathbf{u}_T, \mathbf{y}_T \rangle \in \Big[ \mathbb{E}_{\mathbf{y}}[\langle \mathbf{u}_T, \mathbf{y}_T \rangle] - \sqrt{2\log(2/\epsilon)} \cdot \|\mathbf{u}_T\|_2, \mathbb{E}_{\mathbf{y}}[\langle \mathbf{u}_T, \mathbf{y}_T \rangle] + \sqrt{2\log(2/\epsilon)} \cdot \|\mathbf{u}_T\|_2 \Big].$$

Notice that

$$\mathbb{E}_{y_i}[u_i y_i] = u_i \mathbb{E}[y_i] = u_i\big( (1-\rho)z_i + \rho(1-2\sigma) \big),$$

$$\mathbb{E}_{\mathbf{y}}[\langle \mathbf{u}_T, \mathbf{y}_T \rangle] = (1-\rho)\sum_{i\in T} u_i z_i + \rho(1-2\sigma)\sum_{i\in T} u_i$$

$$= (1-\rho) \cdot \langle \mathbf{w}_T \odot \mathbf{z}_T, \mathbf{x}_T \rangle + \rho(1-2\sigma) \cdot \langle \mathbf{w}_T, \mathbf{x}_T \rangle,$$

since $\mathbf{x}$ is $(\alpha,\lambda)$-strictly subexponential, we have

$$\mathbb{P}_{\mathbf{x}}\Big[ |\langle \mathbf{w}_T, \mathbf{x}_T \rangle| > t \cdot \|\mathbf{w}_T\|_2 \Big] \leq 2\exp\big( -(t/\lambda)^{1+\alpha} \big),$$

and

$$\mathbb{P}_{\mathbf{x}}\Big[ |\langle \mathbf{w}_T \odot \mathbf{z}_T, \mathbf{x}_T \rangle| > t \cdot \|\mathbf{w}_T\|_2 \Big] \leq 2\exp\big( -(t/\lambda)^{1+\alpha} \big).$$

Thus, with probability at least $1-\epsilon$ of $\mathbf{x}$ it holds that

$$|\langle \mathbf{w}_T, \mathbf{x}_T \rangle| \leq \lambda \log^{\frac{1}{1+\alpha}}(4/\epsilon) \cdot \|\mathbf{w}_T\|_2,$$

and

$$|\langle \mathbf{w}_T \odot \mathbf{z}_T, \mathbf{x}_T \rangle| \leq \lambda \log^{\frac{1}{1+\alpha}}(4/\epsilon) \cdot \|\mathbf{w}_T\|_2.$$

Notice that $\mathbf{x}$ is distributed on $\{\pm 1\}^n$ and hence $\|\mathbf{u}_T\|_2 = \|\mathbf{w}_T \odot \mathbf{x}_T\|_2 = \|\mathbf{w}_T\|_2$, then with probability at least $1-\epsilon$ of $\mathbf{y}$ it holds that

$$\begin{aligned}
|\langle \mathbf{u}_T, \mathbf{y}_T \rangle| &\leq \big| \mathbb{E}_{\mathbf{y}}[\langle \mathbf{u}_T, \mathbf{y}_T \rangle] \big| + \sqrt{2\log(2/\epsilon)} \cdot \|\mathbf{u}_T\|_2 \\
&\leq (1-\rho) \cdot |\langle \mathbf{w}_T \odot \mathbf{z}_T, \mathbf{x}_T \rangle| + \rho(1-2\sigma) \cdot |\langle \mathbf{w}_T, \mathbf{x}_T \rangle| + \sqrt{2\log(2/\epsilon)} \cdot \|\mathbf{u}_T\|_2 \\
&\leq \big( (1-\rho) + \rho(1-2\sigma) \big) \cdot \lambda \log^{\frac{1}{1+\alpha}}(4/\epsilon) \cdot \|\mathbf{w}_T\|_2 + \sqrt{2\log(2/\epsilon)} \cdot \|\mathbf{u}_T\|_2 \qquad \text{(C.2)} \\
&\leq (1-2\rho\sigma) \cdot \lambda \log^{\frac{1}{1+\alpha}}(4/\epsilon) \cdot \|\mathbf{w}_T\|_2 + \sqrt{2\log(2/\epsilon)} \cdot \|\mathbf{u}_T\|_2 \\
&= C \cdot \|\mathbf{u}_T\|_2,
\end{aligned}$$

where $C = (1-2\rho\sigma) \cdot \lambda \log^{\frac{1}{1+\alpha}}(4/\epsilon) + \sqrt{2\log(2/\epsilon)}$. ∎

**Lemma C.2** *Suppose that the vector $\mathbf{w} \in \mathbb{R}^n$ is $\alpha$-regular, i.e., $\|\mathbf{w}\|_\infty \leq \alpha \cdot \|\mathbf{w}\|_2$. Suppose $\mathbf{u}$ is a $n$-dimensional random vector where each coordinate is 1 with probability $\rho$ and 0 with probability $1 - \rho$ independently. If $\alpha \leq \rho/\sqrt{\log(1/\delta)/2}$, then with probability at least $1 - \delta$ over the randomness of $\mathbf{u}$ it holds that*

$$\|\mathbf{w}\|_\infty \leq \alpha \cdot \|\mathbf{w}\|_2 \leq c \cdot \alpha \cdot \|\mathbf{w} \odot \mathbf{u}\|_2,$$

*where $c = (\rho - \sqrt{\log(1/\delta)/2} \cdot \alpha)^{-\frac{1}{2}}$. If condition $\alpha \leq \rho/\sqrt{2\log(1/\delta)}$ holds, then with probability at least $1 - \delta$ over the randomness of $\mathbf{u}$ it holds that*

$$\|\mathbf{w}\|_\infty \leq \alpha \cdot \|\mathbf{w}\|_2 \leq (\rho/2)^{-\frac{1}{2}} \cdot \alpha \cdot \|\mathbf{w} \odot \mathbf{u}\|_2.$$

**Proof** By Hoeffding's inequality and notice that $\mathbb{E}[(w_i u_i)^2] = w_i^2 \cdot \mathbb{E}[u_i] = \rho w_i^2$ and $0 \leq (w_i u_i)^2 \leq w_i^2$, we obtain

$$\mathbb{P}\Big( \|\mathbf{w} \odot \mathbf{u}\|_2^2 - \rho \cdot \|\mathbf{w}\|_2^2 \leq -t \Big) \leq \exp\Big( -\frac{2t^2}{\|\mathbf{w}\|_4^4} \Big)$$

$$\leq \exp\Big( -\frac{2t^2}{\|\mathbf{w}\|_\infty^2 \|\mathbf{w}\|_2^2} \Big)$$

$$\leq \exp\Big( -\frac{2t^2}{\alpha^2 \cdot \|\mathbf{w}\|_2^4} \Big),$$

where the second inequality is by the definition of the infinity norm, and the third inequality is by the $\alpha$-regularity condition. Therefore, with probability at least $1 - \delta$, we can get

$$\|\mathbf{w} \odot \mathbf{u}\|_2^2 \geq \rho \cdot \|\mathbf{w}\|_2^2 - \sqrt{\frac{\log(1/\delta)}{2}} \cdot \alpha \cdot \|\mathbf{w}\|_2^2.$$

Then, with probability at least $1 - \delta$, it holds that

$$\|\mathbf{w}\|_\infty \leq \alpha \cdot \|\mathbf{w}\|_2 \leq \alpha \cdot \Big( \rho - \sqrt{\frac{\log(1/\delta)}{2}} \cdot \alpha \Big)^{-\frac{1}{2}} \cdot \|\mathbf{w} \odot \mathbf{u}\|_2.$$

$\blacksquare$

**Theorem C.3 (Berry-Esseen CLT)** *Let $X_1, X_2, ...,$ be independent random variables with $\mathbb{E}[X_i] = 0, \mathbb{E}[X_i^2] = \sigma_i^2 > 0$, and $\mathbb{E}[|X_i|^3] = \rho_i < \infty$. Also, let*

$$S_n = \frac{X_1 + X_2 + \cdots + X_n}{\sqrt{\sigma_1^2 + \sigma_2^2 + \cdots + \sigma_n^2}}$$

*be the normalized $n$-th partial sum. Denote $F_n$ the cdf of $S_n$, and $\Phi$ the cdf of the standard normal distribution. Then for all $n$ there exists an absolute constant $C$ such that*

$$\sup_{x \in \mathbb{R}} |F_n(x) - \Phi(x)| \leq C \cdot \Big( \sum_{i=1}^n \sigma_i^2 \Big)^{-\frac{3}{2}} \cdot \sum_{i=1}^n \rho_i.$$

**Lemma C.4 (Lemma D.11 in Chandrasekaran et al. (2024))** *For $T > 0$ and error $\epsilon > 0$, there exists a polynomial $p$ such that*

1. $\sup_{x \in [0,T]} |p(x) - e^{-x}| \leq \epsilon$.

2. $\deg(p) \leq O(\sqrt{T \log(1/\epsilon)})$, *if* $T = \omega(\log(1/\epsilon))$.

3. $p(x) = \sum_{i=1}^{\deg(p)} c_i x^i$ *where* $|c_i| \leq e^{C\sqrt{T \log(1/\epsilon)}}$ *for all* $i \leq \deg(p)$. *Here $C$ is a large enough constant.*

**Lemma C.5** *If $x$ is $(\alpha, \lambda)$-strictly sub-exponential random variable satisfying $\mathbb{P}_x[|x| > t] \leq 2 \cdot e^{-(t/\lambda)^{1+\alpha}}$, then the $k$-th moment is upper bounded by:*

$$E_x[|x|^k] \leq C\lambda^k \cdot \Big( \frac{k}{1+\alpha} \Big)^{\frac{k}{1+\alpha} + \frac{1}{2}} e^{-\frac{k}{1+\alpha} + \frac{1+\alpha}{12k}},$$

*where $C$ is a universal constant.*

**Proof** By the layer cake representation and the tail bound, we have

$$E_x[|x|^k] = k \int_0^\infty t^{k-1} \cdot \mathbb{P}_x[|x| > t] dt \le 2k \int_0^\infty t^{k-1} \cdot e^{-(t/\lambda)^{1+\alpha}} dt.$$

Making the substitution $s = (t/\lambda)^{1+\alpha}$, i.e. $t = \lambda s^{\frac{1}{1+\alpha}}$ and $dt = \frac{\lambda}{1+\alpha} s^{-\frac{\alpha}{1+\alpha}} ds$, we get

$$E_x[|x|^k] \le 2k \int_0^\infty \lambda^{k-1} s^{\frac{k-1}{1+\alpha}} \cdot e^{-s} \cdot \frac{\lambda}{1+\alpha} s^{-\frac{\alpha}{1+\alpha}} ds$$

$$= \frac{2k\lambda^k}{1+\alpha} \int_0^\infty s^{\frac{k}{1+\alpha}-1} \cdot e^{-s} ds$$

$$= \frac{2k\lambda^k}{1+\alpha} \cdot \Gamma\Big(\frac{k}{1+\alpha}\Big).$$

Notice that $\Gamma(x) \le C x^{x-1/2} e^{-x} e^{1/(12x)}$ for a positive constant $C$, then we have

$$E_x[|x|^k] \le \frac{2k\lambda^k}{1+\alpha} \cdot C\Big(\frac{k}{1+\alpha}\Big)^{\frac{k}{1+\alpha}-\frac{1}{2}} e^{-\frac{k}{1+\alpha}+\frac{1+\alpha}{12k}}.$$

∎

**Lemma C.6** *If $x$ is $(\alpha, \lambda)$-strictly sub-exponential random variable satisfying $\mathbb{P}_x[|x| > t] \le 2 \cdot e^{-(t/\lambda)^{1+\alpha}}$, then for any $a > 0$*

$$\mathbb{E}_x[e^{a|x|}] \le 3 e^{2^{1/\alpha}(a\lambda)^{1+1/\alpha}}.$$

**Proof** We split the integral into two parts at some threshold $T$, which we choose later:

$$\mathbb{E}_x[e^{a|x|}] = \mathbb{E}_x\big[e^{a|x|} \cdot \mathbb{1}[|x| < T]\big] + \mathbb{E}_x\big[e^{a|x|} \cdot \mathbb{1}[|x| \ge T]\big]$$

$$\le e^{aT} + \mathbb{E}_x\big[e^{a|x|} \cdot \mathbb{1}[|x| \ge T]\big].$$

By the layer cake representation and the tail bound, we have

$$\mathbb{E}_x\big[e^{a|x|} \cdot \mathbb{1}[|x| \ge T]\big] = \int_T^\infty a e^{at} \cdot \mathbb{P}_x[|x| > t] dt \le 2a \int_T^\infty e^{at} \cdot e^{-(t/\lambda)^{1+\alpha}} dt.$$

Choose $T$ to be $(2a)^{1/\alpha} \lambda^{1+1/\alpha}$. Then, we have $(t/\lambda)^{1+\alpha} \ge 2at$ for $t \ge T$ and hence

$$\mathbb{E}_x\big[e^{a|x|} \cdot \mathbb{1}[|x| \ge T]\big] \le 2a \int_T^\infty e^{-at} dt = 2e^{-aT}.$$

Thus,

$$\mathbb{E}_x[e^{a|x|}] \le e^{aT} + 2e^{-aT} \le 3e^{aT} = 3e^{2^{1/\alpha}(a\lambda)^{1+1/\alpha}}.$$

∎

