# OpenReview forum: "Smoothed Agnostic Learning of Halfspaces over the Hypercube"
_NeurIPS.cc/2025/Conference — NeurIPS 2025 poster_

### Official Review · Reviewer_PoAm · 2025-07-01

**Clarity:** 4
**Significance:** 3
**Originality:** 3
**Rating:** 5
**Confidence:** 4

**Summary:**

The work studies and extends the smoothed learning setting introduced in [Chandrasekaran et al 2024].
This setting is a relaxation of the agnostic learning model, aiming to sidestep computational hardness results via relaxing the learning objective as follows. In the agnostic learning model, the learning algorithm is required to achieve error $opt+\varepsilon$, where $opt$ is the smallest classification error in the hypothesis class.

In the smoothed learning model of [Chandrasekaran et al 2024] the classifier is only required to compete with hypotheses that are robust to a small amount of Gaussian noise in the input. Formally, the prediction error should be $opt_{smooth}+\varepsilon$, and $opt_{smooth}$ is the smallest **smoothed loss** in the hypothesis class. The smoothed loss of a hypothesis $h$  is defined in [Chandrasekaran et al 2024] as the probability that $h(x+\sigma z)\neq y$ where $(x,y)$ comes from the data distribution and $z$ is the Gaussian noise.

Conceptually, if the smoothed loss of a hypothesis is close to its prediction error, then the hypothesis is robust to noise, and vice versa.
For many hardness results, the optimal classifiers are fragile to small amounts of Gaussian noise, and this new notion of optimality allows [Chandrasekaran et al 2024] to sidestep such hardness results.

The original setting in [Chandrasekaran et al 2024] focuses on continuous domain $\mathbb{R}^d$, whereas this work focuses on distributions over the $n$-dimensional bit strings $\\{\pm 1\\}^n$. The Gaussian perturbations studied in [Chandrasekaran et al 2024] are not appropriate for $\\{\pm 1\\}^n$, because such perturbation will map points in $\\{\pm 1\\}^n$ to points outside of $\\{\pm 1\\}^n$. To remedy this, this paper considers smoothing perturbations that flip each of the bits with probability $\sigma$. This type of perturbation has been widely studied in the context of theory of Boolean functions.

This work gives an algorithm that:
- Makes an assumption that the training marginal is $\sigma_0$-sub-Gaussian. This is a very mild assumption on the tail behavior of the distribution. The work also considers the more general family of $(\alpha, \lambda)$-strictly sub-exponential distributions.
- Considers the hypothesis class of linear classifiers.
- Gives a classifier with prediction error of $opt_{smooth}+\varepsilon$, where $opt_{smooth}$ is defined with respect to Boolean bit flips described above.
- Has run-time is $n^{poly(\frac{\sigma_0}{\sigma \epsilon})}$

The algorithm is based on the low-degree method of [Kalai et al '08]. This algorithm is analyzed by showing that halfspaces are well-approximated by polynomials of degree $poly(\frac{\sigma_0}{\sigma \epsilon})$. The analysis
 involves combining the analysis of [Chandrasekaran et al 2024] with techniques from Critical Index theory. Conceptually, if no single variable has a large weight for the linear classifier, the inner product of the added noise with the normal vector is distributed similar to a Gaussian (via a central limit theorem), and thus methods of [Chandrasekaran et al 2024] can be used. Otherwise, the paper shows that one can use methods similar to [Diakonikolas et al 2010] to separate the high-weight variables and handle these high-weight variables separately.

**Questions:**

- The run-time is $n^{poly(\frac{\sigma_0}{\sigma \epsilon})}$. Is the polynomial factor hidden by $poly$ optimal in some sense?
- One way to think about the technical approach is that it combines the ideas in [Chandrasekaran et al 2024] with the overall approach in [Diakonikolas et al 2010]. To better understand the originality of your approach, could you elaborate on the biggest specific points of difference between your approach and [Diakonikolas et al 2010]?
- Does your work imply distribution-specific agnostic learning algorithms with respect to smoothed distributions? (i.e. distributions formed by starting with an arbitrary sub-Gaussian distribution and adding independent noise). If this is the case, I think this should be highlighted.

**Ethical Concerns:**

["NO or VERY MINOR ethics concerns only"]

**Final Justification:**

Based on the detailed comparison with [Diakonikolas et al] contained in the rebuttal, I think it is appropriate to raise my score to 5 (Accept), based on technical novelty.

**Limitations:**

I think that the paper would benefit from a section titled "Limitations" in which the main limitations are summarized. This would make the limitations clearer to a wider audience.

**Quality:**

4

**Strengths And Weaknesses:**

Strengths
- Smoothed agnostic learning has recently attracted a lot of attention as a promising framework for sidestepping hardness results in theory of machine learning.
- This work extends some of the results in [Chandrasekaran et al 2024] into a natural setting in the Boolean cube, and has the potential to stimulate future extensions for general product distributions.

Weaknesses
- Compared to [Chandrasekaran et al 2024], the scope of this work is more narrow, because this work only considers the class of halfspaces. In contrast, [Chandrasekaran et al 2024] handle a much more general class of functions that includes functions of low Gaussian surface area embedded into a low-dimensional subspace. For example, the class of functions studied in [Chandrasekaran et al 2024] includes intersections of halfspaces, but this work does not handle intersections of halfspaces.

---

> ### Author Rebuttal · Authors · 2025-07-31
>
> We thank the reviewer for the constructive feedback and thoughtful questions. Below, we address each point in turn.
>
> **Q1:** The runtime is $n^{poly(\sigma_0/\sigma\epsilon)}$. Is the polynomial factor hidden by poly optimal in some sense?
>
> **A1:** We thank the reviewer for the thoughtful question. We interpret the question as asking whether the degree of the polynomial hidden in the runtime expression $n^{poly(\sigma_0/\sigma\epsilon)}$ is optimal under our assumptions—or whether it could be improved.
>
> This dependence arises from the degree of the polynomial needed to approximate the smoothed halfspace. Specifically, Lemma 4.7 shows that to achieve \epsilon-approximation under a $(\alpha,\lambda)$-strictly sub-exponential distribution, it suffices to use a polynomial of degree $d=(\frac{C\lambda\log(1/\epsilon)}{\epsilon\sigma^{1/2}})^{\Theta((1+\frac{1}{\alpha})^{3})}$. In the special case of $\sigma_0$-sub-gaussian distributions, this simplifies to $d=(\frac{C\sigma_0\log(1/\epsilon)}{\epsilon\sigma^{1/2}})^{\Theta(1)}$. This degree then drives the runtime via a reduction to L1-regression, yielding the stated complexity.
>
> We do not claim that this exponent is tight in all regimes. In fact, the analysis in Theorem 4.8 is somewhat conservative: tighter control over moments or distribution-specific structure could reduce the required degree, possibly improving the exponent of degree $d$ with a smaller constant factor. However, we believe the polynomial dependencies on $\sigma_0, 1/\sigma, 1/\epsilon$ in the exponent are still necessary in our analysis. In particular, dependence on $1/\sigma$ is unavoidable: as $\sigma\to 0$, our complexity approaches that of standard agnostic learning, which is known to be computationally hard even under the uniform distribution [1]. Thus, the complexity lower bound aligns with known hardness as $\sigma\to 0$, and our result can be seen as a smooth transition from hardness to tractability.
>
> If the reviewer had a different notion of optimality in mind, we’d be happy to clarify further.
>
> **Q2:** One way to think about the technical approach is that it combines the ideas in [2] with the overall approach in [3]. To better understand the originality of your approach, could you elaborate on the biggest specific points of difference between your approach and [3]?
>
> **A2:** We appreciate the reviewer’s insightful comparison between our approach and that of [3]. While both works share high-level similarities—such as the use of low-degree polynomial approximations and critical index-based decomposition—their goals, techniques, and assumptions differ substantially.
>
> **Purpose and setting:** The goal of [3] is to construct pseudorandom generators that fool halfspaces under the uniform distribution using bounded independence. Their focus is on indistinguishability and derandomization. In contrast, our focus is algorithmic: we aim to learn halfspaces efficiently in the smoothed agnostic setting over the Boolean cube, under strictly sub-exponential marginals. Our objective is to design a learner that competes with the smoothed optimum $opt_{\sigma}$, rather than showing indistinguishability under limited independence.
>
> **Differences in use of polynomials:** Both works use low-degree polynomial approximations, and are similar in this regard. However, the way polynomials are constructed is fundamentally different.  In [3], a single polynomial is constructed in a somewhat direct-way by carefully reducing the question to a univariate approximation problem. In our case, we essentially construct a distribution over polynomials (we get a different polynomial $p_z(x)$ for each $z$, and the final approximating polynomial is essentially of the form $E_z[p_z(x)]$).
>
> Further, although both approaches use critical index decomposition to handle irregular weight vectors, the construction of polynomial approximators is different in both goal and implementation.
>
> In the small critical index case, both our work and [3] condition on the “head” variables and analyze the regular “tail.” In [3], after fixing the head, the tail defines an $\epsilon$-regular halfspace under the uniform distribution. The authors show that such functions can be fooled by bounded-independence distributions, using sandwiching polynomials that tightly bound the target function from above and below. These sandwiching polynomials are constructed by reduction to a univariate problem.
>
> In contrast, after conditioning on the head, we analyze the function $f(x \odot z)$, and approximate its smoothed version $T_{1-\rho} f_x(z)$. The tail remains a regular Boolean halfspace, which we approximate by a Gaussian via the Berry–Esseen theorem. This enables the use of the density ratio (probability tilting) technique from [2] to approximate the expected value of the smoothed function with a low-degree polynomial. Importantly, because the head is small and fixed in this step, the final approximator remains low-degree in the Boolean domain, making it suitable for L1 regression. Note that after conditioning on the head, our approach differs from [3].
>
> For the large critical index case, both works use the idea that the head dominates the decision. In [3], this is used to argue that the head variables give sufficient bias to fool the function even under pairwise independence. In our setting, the dominance of the head allows us to exactly represent the function as a low-degree Boolean polynomial, which can then be directly used in regression.
>
> **Distributional assumptions:** [3] assumes the uniform distribution over $\\{\pm1\\}^n$. In contrast, our results hold under strictly sub-exponential marginals, which are significantly more general. This generality introduces new technical challenges—for instance, ensuring that regularity and Gaussian approximation continue to hold under dependent noise. We address these using smoothing, critical index decomposition, and novel tools such as discrete noise operators and rerandomization techniques.
>
> While we are inspired by the structural decomposition used in [3], our work addresses a fundamentally different problem. Our goals (agnostic learning vs. pseudorandomness), approximation objectives (expected L1 error vs. pointwise bounds), and assumptions (general sub-exponential marginals vs. uniform distribution) all differ substantially. Moreover, our analysis introduces new techniques tailored to the smoothed Boolean setting. We believe these differences position our work as a distinct contribution to the study of efficient learning under noise and distributional uncertainty.
>
> **Q3:** Does your work imply distribution-specific agnostic learning algorithms with respect to smoothed distributions? (i.e. distributions formed by starting with an arbitrary sub-Gaussian distribution and adding independent noise). If this is the case, I think this should be highlighted.
>
> **A3:** We appreciate the reviewer’s suggestion and the opportunity to clarify the connection to agnostic learning under smoothed distributions. Indeed, it is possible that smoothed-agnostic model is strictly more general than learning under smoothed distributions as happens in the Gaussian case (e.g., as in [2]). However, this could be a bit trickier to do in the Boolean case and we leave this as an interesting extension for future work. We will revise the paper to highlight this direction.
>
> **Q4:** I think that the paper would benefit from a section titled "Limitations" in which the main limitations are summarized. This would make the limitations clearer to a wider audience.
>
> **A4:** We thank the reviewer for the constructive suggestion regarding a clearer discussion of limitations. We agree that summarizing the main limitations in a dedicated section will improve accessibility for a broader audience. In the camera-ready version, we will add a new “Limitations” section immediately preceding the Conclusion and Future Work. This section will explicitly highlight key constraints of our approach—including the restriction to halfspaces, the dependence of runtime on $\sigma$, the necessity of tail decay assumptions, and the current focus on Boolean inputs with bit-flip noise. We will also revise the organization of the final sections accordingly for better clarity.
>
> **References:**
>
> [1] Kalai, Adam Tauman, et al. "Agnostically learning halfspaces." SIAM Journal on Computing 37.6 (2008): 1777-1805.
>
> [2] Chandrasekaran, Gautam, et al. "Smoothed analysis for learning concepts with low intrinsic dimension." COLT 2024.
>
> [3] Diakonikolas, Ilias, et al. "Bounded independence fools halfspaces." SIAM Journal on Computing 39.8 (2010): 3441-3462.
>
> [4] Kane, Daniel, Adam Klivans, and Raghu Meka. "Learning halfspaces under log-concave densities: Polynomial approximations and moment matching." COLT 2013.

---

> > ### Comment · Reviewer_PoAm · 2025-08-04
> >
> > Thank you for your comments. I confirm that your interpretation of Q1 was correct (sorry for possible ambiguity).
> > If the paper is accepted, I think it is appropriate to include the answers you gave above in the final version.
> >
> > Based on the detailed comparison with [Diakonikolas et al], I think it is appropriate to raise my score to 5 (Accept), based on technical novelty.

---

### Official Review · Reviewer_zBEU · 2025-07-02

**Clarity:** 1
**Significance:** 2
**Originality:** 2
**Rating:** 5
**Confidence:** 1

**Summary:**

This paper considers the setting of smoothed agnostic learning halfspace on a hypercube.
This setting differs from the previous work (CKKM24) in the sense that the previous work considers learning on \R^d, while this one considers learning on the hypercube.
Namely, the setting considered here is the following:
Given a sample access to a joint distribution (x,y) over {\pm 1}^n\times {\pm 1}, the algorithm need to output a hypothesis with error small
compared to the smoothed agnostic error,
defined in Definition 1.2.

Given the setting, further assume that the distribution is $(\alpha,\lambda)$-subexponential (Definition 4.6), the algorithms run in time
$n^{poly({(\lambda/(\sigma\epsilon))}^{{(1+1/\alpha)}^3}}$ and learns to error opt+\eps where opt is the smoothed optimal error above.
The high-level approach here follows the L1 regression algorithm framework from KKMS08, which boils down to showing that the concept function can be approximated by a polynomial of corresponding degree to small L1 norm (Lemma 4.7).
The notion of approximation is defined in Lemma 4.2.
I'm then having trouble following the authors' argument.
In particular, in lines 214-223:
216 states that the approximation polynomial (as a function of x) needs to be the same for every z (which is consistent with Lemma 4.7 and 4.2).
Then 221 states that the polynomial (now as a function of z) needs to approximate the function f_x(z).
I assume that this means now this polynomial needs to be fixed for every x to approximate a function on z, which is inconsistent with the above and it wan't clear to me that one would imply the other.

**Questions:**

Can the authors clarify my question about 214-223 (as I mentioned above)?
I'm happy to revise the review if I misunderstood it.

**Ethical Concerns:**

["NO or VERY MINOR ethics concerns only"]

**Final Justification:**

The authors have clarified my confusions.

**Limitations:**

Yes.

**Quality:**

2

**Strengths And Weaknesses:**

Weaknesses:
Some parts of the presentation are confusing, especially lines 214-223, which I mentioned earlier.

---

> ### Author Rebuttal · Authors · 2025-07-24
>
> **Q:** Given the setting, I’m having trouble following the authors' argument between lines 214–223. Line 216 suggests that the approximation polynomial (as a function of $x$) needs to be the same for every $z$, which matches Lemmas 4.2 and 4.7. But line 221 seems to suggest constructing a polynomial in $z$ to approximate $f_x(z)$, which would require fixing $x$ instead. This appears inconsistent, and it's unclear how one implies the other.
>
> **A:** The confusion arises from a misinterpretation of the roles of $x$, and $z$ in our construction.
>
> To clarify, our goal, as stated in Theorem 4.2, is to construct a family of low-degree polynomials $\{p_z(x)\}$—each of which is a polynomial in $x$ for a fixed noise vector $z$. There is no requirement that a single polynomial in $x$ must work for all $z$, nor that $p_z(x)$ be a polynomial in $z$ when $x$ is fixed. If $p_z(x)$ is a polynomial in $x$ for each $z$, then the sum of such polynomials over $z$ would also be a polynomial.
>
> To analyze and construct such polynomials, we temporarily fix $x$ and define $f_x(z) := f(x \odot z)$, which allows us to study the smoothed function via the operator $T_{1-\rho} f_x(z)$ acting on $z$. This step is purely analytical: it enables the use of tools like the Berry–Esseen theorem, which apply to functions over random noise vectors. This perspective—treating $f_x(z)$ as a function of $z$—helps us approximate the expected value of $f(x \odot z)$, which in turn determines the value of the polynomial $p_z(x)$.
>
> To be clear: we do not construct a polynomial in $z$. Instead, the role of $z$ is to index the family of polynomials $\{p_z(x)\}$; for each fixed $z$, we define a corresponding polynomial $p_z(x)$ in $x$, and this is what is ultimately used in the L1 regression algorithm. The approximation argument involving $f_x(z)$ and $T_{1-\rho}$ is a technical tool that helps derive these $x$-polynomials.
>
> This shift in viewpoint is subtle but is important. We will revise this part to more clearly underscore this shift and explain the purpose of introducing $f_x(z)$.
>
> Thank you again for highlighting this, and we welcome further questions.

---

> > ### Comment · Reviewer_zBEU · 2025-08-04
> >
> > Thanks for all the detailed clarifications and answers. I will raise my score.

---

### Official Review · Reviewer_Mp29 · 2025-07-04

**Clarity:** 4
**Significance:** 3
**Originality:** 4
**Rating:** 5
**Confidence:** 4

**Summary:**

This work studied the problem of agnostic learning halfspaces with Boolean marginal under the smoothed agnostic learning setting. Interestingly, the authors showed that, the target halfspace can be learned (in the sense of $\sigma$-smoothed agnostic learning) under very general $\sigma_0$-sub-gaussian marginals on the boolean hypercube using $O(n^{poly(\sigma_0/\sigma\epsilon)})$ samples. Prior work often requires strong assumptions of the marginal distribution, e.g., the marginal is a product distribution etc. The main idea is to replace the target halfspace with its smoothed version, and then approximate the smoothed halfspace using low-degree polynomials. Important techniques includes an analysis of the weights using critical index and applying Berry-Essen to invoke continuous tools.

**Questions:**

I wonder is there an upper bound on the 0-1 loss of the output hypothesis?

**Ethical Concerns:**

["NO or VERY MINOR ethics concerns only"]

**Final Justification:**

I recommend acceptance.

**Limitations:**

no negative societal limitations.

**Paper Formatting Concerns:**

no concerns

**Quality:**

4

**Strengths And Weaknesses:**

1. The paper is written in a very clear and fluent way, with many explanations and intuitions.
2. The paper is strong and solid in techniques.
3. It is interesting to see that it is possible to learn halfspaces agnostically under the smoothed analysis setting.

---

> ### Author Rebuttal · Authors · 2025-07-31
>
> **Q:** I wonder if there is an upper bound on the 0-1 loss of the output hypothesis?
>
> **A:** We thank the reviewer for the encouraging feedback and for raising this thoughtful question.
>
> Our algorithm’s guarantee is with respect to the smoothed agnostic error—i.e., it outputs a hypothesis $h$ such that $P_{(x,y)\sim D}[h(x)\neq y]\leq opt_{\sigma}+\epsilon$, where $opt_{\sigma}$ denotes the minimum achievable error by any halfspace evaluated on $\sigma$-bit-flip-perturbed examples. This is consistent with the goal of smoothed analysis, which seeks to evaluate performance on slightly perturbed instances to mitigate worst-case pathologies.
>
> As for the true 0–1 loss of the output hypothesis on unperturbed examples, we do not provide an upper bound in terms of the standard agnostic error $opt_0$, since doing so would contradict known hardness results. In particular, prior work [1] shows that agnostically learning halfspaces over the Boolean cube is computationally hard, even under simple distributions like the uniform distribution. Therefore, without further assumptions, we cannot hope to relate the true 0–1 loss of our hypothesis to the best achievable 0–1 loss in the original distribution.
>
> That said, in practice, the output hypothesis may still achieve low 0–1 error under benign distributions—especially if the optimal halfspace is stable under small perturbations. Our result ensures that, under bit-flip noise and sub-exponential marginals, the learner competes with the smoothed optimum efficiently.
>
> We hope this clarifies the scope and motivation of our guarantees.
>
> **References:**
>
> [1] Kalai, Adam Tauman, et al. "Agnostically learning halfspaces." SIAM Journal on Computing 37.6 (2008): 1777-1805.

---

### Official Review · Reviewer_PZPU · 2025-07-08

**Clarity:** 2
**Significance:** 3
**Originality:** 2
**Rating:** 5
**Confidence:** 3

**Summary:**

This paper studies agnostic PAC learning of halfspaces over the Boolean hypercube. Without further assumption this problem is hard. The authors tackle the smoothed setting, recently introduced  introduced by Chandrasekaran et al. (2024), which is based on smoothed analysis from computational complexity. Here the goal is not too output a predictor whose true loss is close to the best true loss in class (i.e., over all halfspaces), but instead the true less of the output predictor should be small compared to the best predictor in class over a slightly *perturbed* instance. This makes the analysis less worst-case. Additionally they assume subgaussian (or somewhat more general strictly subexponential) marginal distributions. While Chandrasekaran et al. (2024) study halfspaces in $R^d$ with Gaussian perturrbations here instead halfspaces over $\\{0,1\\}^n$ are studied (with bit flips).This requires a new perturbation model and various adaptations of previous techniques to achieve a sample complexity and runtime of roughly $n^{\mathrm{poly}(\cdot)}$. Relying on Kalai et al. (2008) they reduce linear classification problem to a $L_1$ polynomial regression problem. By replacing the target function with a smooth surrogate and further technical ingredients (from the analysis of Boolean halfspaces) they end up with two cases: either the normal vector is dominated by a small number of components (which can be estimated directly) or the CDF can be approximated well enough through the Berry-Esseen theorem (and thus the problem is reduced to the Gaussian case of Chandrasekaran et al. (2024)).

**Questions:**

Perhaps a somewhat naive question about Def. 4.6. on $(\alpha,\lambda)$-strictly sub-exponential distributions: Isn't any distribution defined over $\\{-1,1\\}^n$ strictly subexponential for some appropriate $(\alpha,\lambda)$? As the domain is bounded it should even be subgaussian? In particular for $t\ge 2\sqrt n$, or so, the tail probability is simply zero. Or would we get a bad dependence on $n$ through that argument. Please clarify.

**Ethical Concerns:**

["NO or VERY MINOR ethics concerns only"]

**Final Justification:**

Authors addressed my concerns. This is a strong paper and should be accepted.

**Limitations:**

yes.

**Paper Formatting Concerns:**

no.

**Quality:**

3

**Strengths And Weaknesses:**

This is a well written, interesting, and technical paper continuing an important line of work. The smoothed agnostic case is well motivated and this paper nicely transfers the results of Chandrasekaran et al. (2024) from $R^d$ to the Boolean case.

Moreover, this paper (by going from worst-case to smoothed/average case) allows substantially more general distributions over the hypercube than were previously studied, such as uniform or product distributions.

The main weakness is the following. **Once addressed I am more than happy to raise my score**. What I am missing is a broader discussion on hardness, tightness, and, in general, necessity/a justification of the various assumptions. For example
* It is discussed (in the related work and in other places) that agnostically learning halfspaces over the Boolean cube is hard, making various distributional and related assumptions justified. However it is not clear why these hardness results remain valid in this new smoothed setting. Perhaps in the smoothed setting learning with arbitrary distributions is indeed not hard? This is not clear to me from the submission.
* From PAC theory we get that $O(n/\epsilon^2)$ samples suffice roughly (VC is $n+1$). Is it clear that a sample complexity/runtime of $O(\mathrm{poly}(n))$ is hard in this smoothed case? This would make the achieved results here—a sample complexity/runtime of $O(n^{\mathrm{poly}(\cdot)})$—more justified.
* In line 201 it is claimed that arbitrary halfspaces might not be well approximated with a low degree polynomial in the worst-case. While this is true in Chandrasekaran et al. (2024) for $R^d$ it is unclear whether this is still true Boolean cube case.

Please correct me if I missed something above/or am wrong.

---

> ### Author Rebuttal · Authors · 2025-07-31
>
> **Q1:** It is discussed that agnostically learning halfspaces over the Boolean cube is hard, making various distributional and related assumptions justified. However it is not clear why these hardness results remain valid in this new smoothed setting. Perhaps in the smoothed setting learning with arbitrary distributions is indeed not hard?
>
> **A1:** We thank the reviewer for this insightful question. Our goal is indeed to provide a positive result in the smoothed setting, but we do not claim that smoothing automatically eliminates all computational hardness. Rather, we identify natural and sufficient conditions—namely, bit-flip noise and strictly sub-exponential marginals—under which agnostic learning becomes tractable.
>
> 1. **Hardness remains as $\sigma\to0$:** Our algorithm runs in time $n^{poly(\sigma_0/\sigma\epsilon)}$, where $\sigma$ is the bit-flip noise rate and $\sigma_0$ is a distribution-dependent constant. Further, it is easy to deduce from existing agnostic learning results that some exponential dependence on sigma is necessary (for example, $\sigma\sim1/poly(n)$ corresponds essentially to no perturbation; at a more fine-grained level, conditional lower bounds of the form $n^{\tilde{\Omega}(1/\epsilon^2)}$ for agnostic learning under the uniform distribution work by reducing to sparse parity with noise on roughly $1/\epsilon^2$ variables and here taking $\sigma\ll1/\epsilon^2$ will reduce perturbations with noise rate $\sigma$ to no noise.) Thus, our result interpolates between the intractable agnostic case and a tractable smoothed setting, rather than bypassing known barriers entirely. We will revise the paper to add this observation.
>
> 2. **Smoothing may mitigate but not necessarily eliminate worst-case structure:** Bit-flip noise introduces independent perturbations to each coordinate, and this smoothing can help regularize the learning problem. However, it remains unclear whether smoothing alone suffices to guarantee low-degree polynomial approximability for arbitrary base distributions. For example, a base distribution concentrated on a low-dimensional subcube or with heavy tails in certain directions may still retain problematic structure even after bit-flip noise. Our current analysis therefore assumes a strict sub-exponential tail condition to ensure sufficient concentration in all directions. Whether this tail decay assumption is necessary or can be removed remains an interesting open question.
>
> 3. **Analogy with smoothed analysis in TCS:** Our framework is consistent with the broader tradition of smoothed analysis [2], where worst-case hardness is overcome under mild random perturbations plus structural conditions. For example, the simplex method remains exponential in the worst case, yet becomes polynomial under Gaussian perturbations. Similarly, our result shows that bit-flip smoothing and tail decay are sufficient (but not excessive) assumptions to enable tractable learning.
>
> **Q2:** From PAC theory we get that $O(n/\epsilon^2)$ samples suffice roughly (VC is $n+1$). Is it clear that a sample complexity/runtime of $O(\mathrm{poly}(n))$ is hard in this smoothed case?
>
> **A2:** It is indeed correct that the sample complexity of agnostically learning halfspaces is $O(n/\epsilon^2)$ due to the VC dimension. However, this is an information-theoretic guarantee — it does not imply the existence of an efficient (i.e., polynomial-time) algorithm that achieves this rate.
>
> Even under the uniform distribution, which satisfies our assumptions, computationally efficient agnostic learning is known to be conditionally hard [1]. Therefore, while a polynomial sample size suffices in principle, achieving polynomial runtime remains a nontrivial challenge without further assumptions.
>
> Our contribution is to identify conditions — namely, bit-flip noise and strictly sub-exponential marginals — under which agnostic learning becomes not only statistically feasible (which it always was), but also computationally efficient. This situates our work within the broader theme of beyond-worst-case analysis, where additional structure is leveraged to overcome known computational barriers.
>
> **Q3:** In line 201 it is claimed that arbitrary halfspaces might not be well approximated with a low degree polynomial in the worst-case. While this is true in Chandrasekaran et al. (2024) for $R^d$ it is unclear whether this is still true Boolean cube case.
>
> **A3:** We thank the reviewer for raising this excellent point. The claim in line 201—that arbitrary halfspaces may not admit good low-degree polynomial approximations in the worst case—does indeed extend to the Boolean cube.
>
> While every Boolean function can be exactly represented by a degree-$n$ polynomial, the key issue is whether low-degree polynomials can approximate arbitrary halfspaces under general distributions. For instance, under the uniform distribution, various tools (e.g., hypercontractivity and Fourier concentration) enable low-degree approximations of regular halfspaces [1]. Under product distributions, spectral concentration results have been used to obtain low-degree approximators [3].
>
> However, these techniques rely on significant structure—uniformity, independence, or symmetry—which may be absent in general sub-exponential distributions. In such settings, standard polynomial approximation techniques (e.g., based on orthogonal polynomials or moment bounds) break down due to uncontrolled tail behavior or dependence.
>
> This motivates our use of smoothed analysis: by applying bit-flip noise, we effectively regularize the target function. The resulting smoothed halfspace admits good low-degree approximations after conditioning and re-randomization, even under unstructured sub-exponential marginals. Thus, the statement in line 201 reflects this fundamental limitation of worst-case polynomial approximation on the Boolean cube, which our smoothing strategy is designed to overcome.
>
> **Q4:** Isn’t any distribution over $\\{\pm 1\\}^n$ strictly sub-exponential for some approximate $\lambda$? As the domain is bounded it should even be subgaussian? In particular for $\lambda=1$, or so, the tail probability is simply zero. Or would we get a bad dependence on $\lambda$ through that argument?
>
> **A4:** While it is true that distributions over $\\{\pm1\\}^n$ have bounded support, and hence projections $\langle w, x \rangle$ are bounded in $[-\\|w\\|_1, \\|w\\|_1] \subseteq [-\sqrt{n}, \sqrt{n}]$, our assumption in Definition 4.6 is stronger than mere boundedness, and it plays a crucial role in enabling low-degree polynomial approximation. In particular, as the reviewer says we would get bad dependence on $\lambda$ (and as a result exponential dependence on the dimension) if we just use boundedness.
> In particular, our proof relies on controlling the low-order moments of $\langle w, x \rangle$ under the input distribution to bound the error of Taylor expansions of smooth surrogates. The $(\lambda, \alpha)$-strictly sub-exponential condition ensures moment growth is controlled in all directions and in a way that is independent of the ambient dimension $n$. This is critical to obtaining an $O(\epsilon)$ polynomial approximation error.
>
> To clarify, the uniform distribution over $\\{\pm1\\}^n$ satisfies the $(\lambda, \alpha)$-strictly sub-exponential condition with $\lambda = 2, \alpha = 1$, using Hoeffding’s inequality. This covers the regime $t = O(\log(1/\epsilon))\ll\sqrt{n}$, where our approximation guarantees operate. We will clarify this in the final version and include the uniform distribution as a canonical example.
> Lastly, our condition subsumes all sub-Gaussian distributions (corresponding to $\alpha = 1$) and excludes those with only linear tail decay $(\alpha = 0)$. This includes common marginals such as unbiased or biased product distributions on $\\{\pm1\\}^n$, making the assumption both meaningful and practically relevant.
>
> **References:**
>
> [1] Kalai, Adam Tauman, et al. "Agnostically learning halfspaces." SIAM Journal on Computing 37.6 (2008): 1777-1805.
>
> [2] Spielman, Daniel, and Shang-Hua Teng. "Smoothed analysis of algorithms: Why the simplex algorithm usually takes polynomial time." STOC. 2001.
>
> [3] BLAIS, E., O’DONNELL, R. and WIMMER, K. (2010). Polynomial regression under arbitrary product distributions. Machine learning 80 273–294.

---

> > ### Comment · Reviewer_PZPU · 2025-08-03
> >
> > Thanks for all the detailed clarifications and answers. I will raise my score and think that paper should be accepted.

---

### Decision · Program_Chairs · 2025-09-17

**Decision:**

Accept (poster)

**Comment:**

This paper studies the problem of agnostically learning halfspaces on the hypercube under a natural class of distributions in a smoothed setting. The authors provide a dimension-efficient algorithm for this learning task. The reviewers agreed that this is a technically non-trivial and interesting contribution that meets the bar for the conference.